# Environmental context-dependent activation of dopamine neurons via putative amygdala-nigra pathway in macaques

Kazutaka Maeda [1,2] ✉, Ken-ichi Inoue [3], Masahiko Takada[3] & Okihide Hikosaka [1] ✉

Seeking out good and avoiding bad objects is critical for survival. In practice, objects are rarely good every time or everywhere, but only at the right time or place. Whereas the basal ganglia (BG) are known to mediate goal-directed behavior, for example, saccades to rewarding objects, it remains unclear how such simple behaviors are rendered contingent on higher-order factors, including environmental context. Here we show that amygdala neurons are sensitive to environments and may regulate putative dopamine (DA) neurons via an inhibitory projection to the substantia nigra (SN). In male macaques, we combined optogenetics with multi-channel recording to demonstrate that rewarding environments induce tonic firing changes in DA neurons as well as phasic responses to rewarding events. These responses may be mediated by disinhibition via a GABAergic projection onto DA neurons, which in turn is suppressed by an inhibitory projection from the amygdala. Thus, the amygdala may provide an additional source of learning to BG circuits, namely contingencies imposed by the environment.

We choose actions in accordance with our goals, for instance to seek out good (rewarding or useful) objects and avoid bad (harmful or useless) objects. Goal-directed actions commonly begin with eye movements towards good objects or away from bad objects. How is such goal-directed behavior controlled by the brain? Eye movements associated with approach and avoidance are controlled by two parallel pathways in basal ganglia (BG) circuits[1,2]. Approach behavior is mediated by the direct pathway from the caudate nucleus (CD) to the superior colliculus (SC) via the substantia nigra pars reticulata (SNr), whereas avoidance behavior is mediated by the indirect pathway through the globus pallidus externus (GPe), i.e., the CD-GPe-SNr-SC pathway. These direct and indirect pathways originate from the caudate head (CDh) and the caudate tail (CDt), which process short- or long–term memory, respectively[3,4].

In real life, different objects are typically found in different environments. Thus, the act of encountering a new environment itself conveys predictive information about what objects are likely to be found therein and accordingly influences behavior[5–7]. How does the brain process such predictive environmental information? We hypothesized that brain regions are likely to play roles that both encode distinct environments and influence the BG circuits. The amygdala is a key structure for encoding emotionally significant environments[6,8–14]. Furthermore, the amygdala sends output to BG, including CDt, GPe, and SNr[15–19]. In the present study, we investigated what information is conveyed through the amygdala-nigra projection using multi-channel recording probes combined with pathway-selective optogenetic technique.

[1]Laboratory of Sensorimotor Research, National Eye Institute, National Institutes of Health, Bethesda, MD 20892, USA. [2]Department of Neurophysiology, National Institute of Neuroscience, National Center of Neurology and Psychiatry, Kodaira, Tokyo 187-8502, Japan. [3]Systems Neuroscience Section, Department of Neuroscience, Primate Research Institute, and Center for the Evolutionary Origins of Human Behavior, Kyoto University, Inuyama, Aichi 484-8506, Japan. ✉e-mail: kaz.maeda.86@gmail.com; oh@lsr.nei.nih.gov

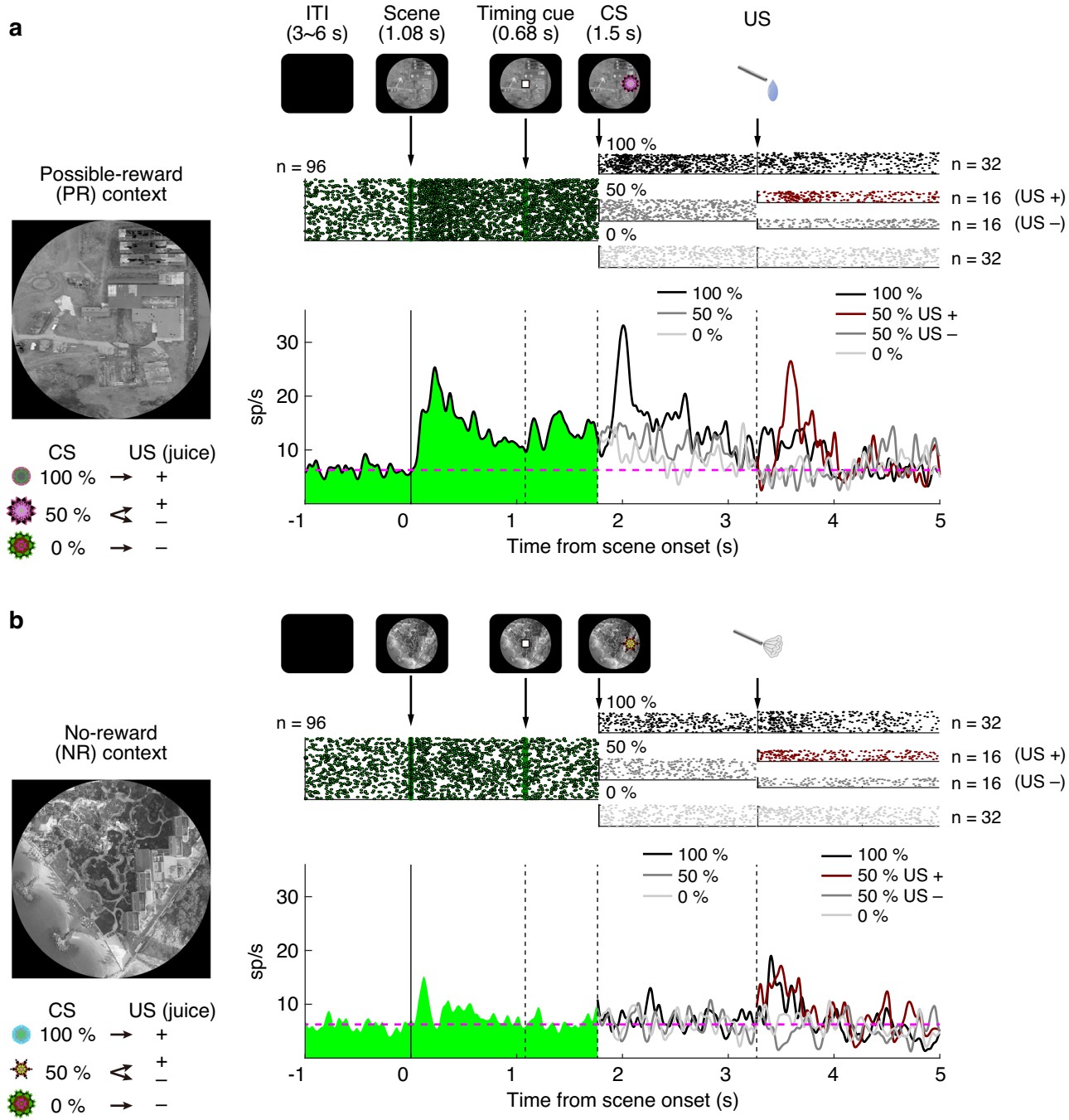

**Fig. 1 | Putative DA response to environmental contexts. a** Responses of a single putative DA neuron in the PR environment of Pavlovian task with scene. In the PR environment, a particular scene was presented, and three conditioned stimuli were associated with water with 100%, 50% and 0% probability, respectively. **b** Responses of the same putative DA neuron in the NR environment of Pavlovian task with scene. In the NR environment, another scene was presented, and the other three conditioned stimuli were associated with air-puff with 100%, 50% and 0% probability, respectively. The environments were pseudo-randomly switched within the block. The trial started after the presentation of a scene image (diameter: 50 degrees) on the screen. After 1.08 s, a timing cue was presented on the scene. After 0.68 s, the timing cue disappeared, and one of the three conditioned stimuli was presented in each environment. After 1.5 s, the conditioned stimulus disappeared, and the unconditioned stimulus (water reward in PR or air-puff in NR) was delivered. Actual scenes and fractals for CSs used are shown on the left. The top row shows a raster plot of firing for the scene and each CS and US in the set, with dots indicating spike times. The bottom row shows the average firing rate in the set. The dotted line indicates the same level as the averaged baseline firing rates. ITI inter-trial interval, PR possible-reward, NR no-reward, CS conditioned stimulus; US, unconditioned stimulus, DA Dopamine.

## Results

### Dopamine activity for environmental contexts

Pavlovian conditioning procedures that pair conditioned stimuli (CS) with unconditioned stimuli (US) have widely been used for testing dopamine (DA) neurons[20]. We designed a Pavlovian procedure in which each trial started with the onset of a distinct environment that constrained what subsequent CS objects and US events could occur. In possible reward (PR) environments, the US took the form of a liquid reward, the occurrence of which was not guaranteed (Fig. 1a). By contrast, no reward (NR) environments signaled the possibility of an air-puff (Fig. 1b). In both environments, the CSs were represented by unique fractal objects associated with their corresponding US with

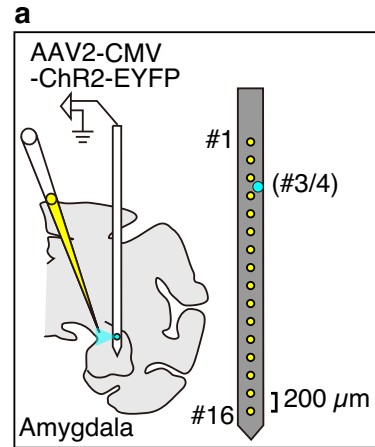
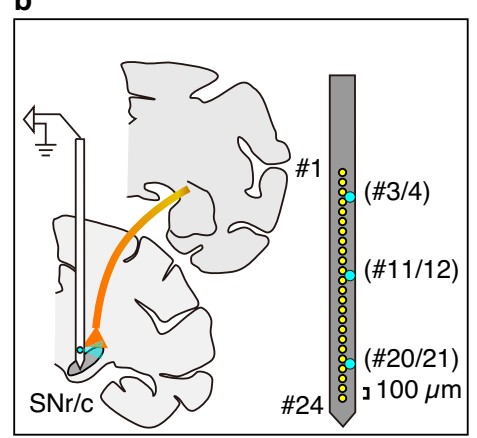

**Fig. 2 | Optogenetic stimulation of amygdala-nigra pathway. a** Injection of the viral vector (AAV2-CMV-ChR2-EYFP) into the amygdala and stimulation/recording of amygdala neurons via a 16-contact probe (yellow dots, 200 μm interelectrode spacing) with a fiber optic which is placed between contact 3 and 4 (cyan dot). **b** Stimulation of ChR2 expressing amygdala axon terminals in the substantia nigra via a 24-contact probe (yellow dots 100 μm interelectrode spacing) with three fiber optics (placed between contacts 3 and 4, contact 11 and 12, and contact 20 and 21; cyan dots).

100%, 50%, or 0%. Fractal CS objects appeared 1.8 s after environment onset. After 1 s, the US occurred (or were omitted) and simultaneously the CS and the environment background image disappeared.

After the monkey learned the task, we recorded single-unit activity from the substantia nigra pars compacta (SNc). When the PR-environment appeared, putative DA neurons showed a phasic response followed by tonic excitation (Fig. 1a). On the other hand, no tonic response was observed in the NR-environment, but putative DA neurons responded to CS and US phasically (Fig. 1b). In the PR-environment, the response to CS was largest when reward prediction was largest (100%), but the response to US was largest when reward outcome was uncertain (50 %). This pattern is typical of DA responses to CS in standard Pavlovian tasks, that is, the absence of a context-predicting cue[21,22]. These data suggest that DA neurons integrate two kinds of information: Environment in the tonic response and Object in the phasic response.

### Amygdala-nigra projection and DA neurons

We hypothesized that the amygdala is a source of the information about environment. Recent studies showed that amygdala neurons send their axons to GABAergic neurons in SNr and modulate neuronal responses therein[12,23]. To study whether DA neurons also receive amygdala input, we injected AAV vector (AAV2-CMV-ChR2-EYFP) into the amygdala, mostly in the central nucleus (CeA) (Fig. 2a) to drive ChR2 expression in amygdala neurons and their axons terminating within the nigra (Fig. 2b). If DA neurons receive input from the amygdala, optical stimulation (O-stim) in the nigra would inhibit or excite DA neurons. For both electrophysiological recording and O-stim, we used a multi-site linear electrode with optic fiber ports inserted in either the amygdala (Fig. 2a) or the nigra (Fig. 2b). The nigra array had three O-stim sites, permitting O-stim at variable distances from the recording contacts (Fig. 2b). Then, we recorded the activity of many neurons while optically stimulating (duration: 200 ms) at one position in each trial. The stimulation intensity was set low enough to avoid irrelevant effects by O-stim (see Methods and Fig. S1). For the amygdala electrode, the effect of O-stim would mostly act on the cell somata of amygdala neurons, whereas for the electrode in the nigra, the effect of O-stim would excite the axon terminals of amygdala neurons.

Many amygdala neurons were modulated by O-stim in the amygdala ($n = 76/211$, 36%), mostly with excitatory responses ($n = 53/76$, 70%, Fig. 3a-left, Fig. 3b: red dots), confirming that excitatory opsin was successfully expressed locally. Some neurons were inhibited ($n = 23/76$, 30%, Fig. 3a-right, Fig. 3b: blue dots), which might not be

due to the direct effect of O-stim on these cells, but was presumably driven by the local inhibitory network of ChR2-expressing neurons. Consistent with this interpretation, inhibited neurons were less tightly clustered around the stimulation site than excited neurons (Fig. S2d), and inhibition occurred at longer latency than excitation (Fig. 3a-right).

Next, we identified putative GABAergic neurons in the substantia nigra (SN) with >15 sp/s baseline activity and phasic inhibition to visual stimuli. We also identified putative DA neurons that had <10 sp/s baseline activity and showed strong excitation to visual stimuli and to 100% reward CS (see Methods for details). Then, we optically stimulated amygdala axons in SN while recording putative GABAergic neurons and putative DA neurons in SN (Fig. 2b). In terms of the characteristic of DA neurons, we measured their spike shape and compared them with that of putative GABAergic neurons. The data showed that the spike shape of putative DA neurons was significantly wider than that of putative GABAergic neurons (Fig. S3). This is consistent with previous data[22]. Most of the affected putative GABAergic neurons exhibited inhibitory responses ($n = 25/35$, 71%, Fig. 2e-left), attributable to monosynaptic projections from GABAergic CeA neurons[24]. Inhibited neurons tended to be found on channels closest to the stimulation port (Fig. 3c-left, Fig. 3d, blue dots). Switching to a different stimulation port led to responses in a new population of nearby neurons (Fig. 3d, green dot line). O-stim drove excited responses in a minority of putative GABAergic neurons (10/35, 29%, Fig. 3c-right, Fig. 3d, red dots), which tended to be weaker in magnitude (Fig. S2b). This pattern of responses could be explained either by a small number of excitatory projections from the amygdala (areas adjacent to CeA), or by disinhibition resulting from disynaptic activation of putative GABAergic neurons. In contrast to putative GABAergic neurons, most putative DA neurons were excited by O-stim (10/12, 83%, Fig. 3e-left), and responsive channels were observed irrespective of the proximity to stimulation ports (Fig. 3f, red dots). This spatial pattern suggests that the effect of O-stim on each putative GABAergic neuron is highly localized to the axon terminals of amygdala neurons that have synaptic inputs to the putative GABAergic neuron. By contrast, the effect of O-stim on each putative DA neuron is spatially variable because the disynaptic inputs from putative GABAergic neurons depend on the spatial variation of their axons.

Further insight into circuit organization can be gleaned by assessing the latency of neuronal responses evoked by O-stim. Putative GABAergic neurons were activated at shorter latency than putative DA neurons (Fig. 4a), again suggesting that putative GABAergic neurons are the primary target of O-stim. Moreover, latency varied systematically as a function of the distance from the stimulation port for

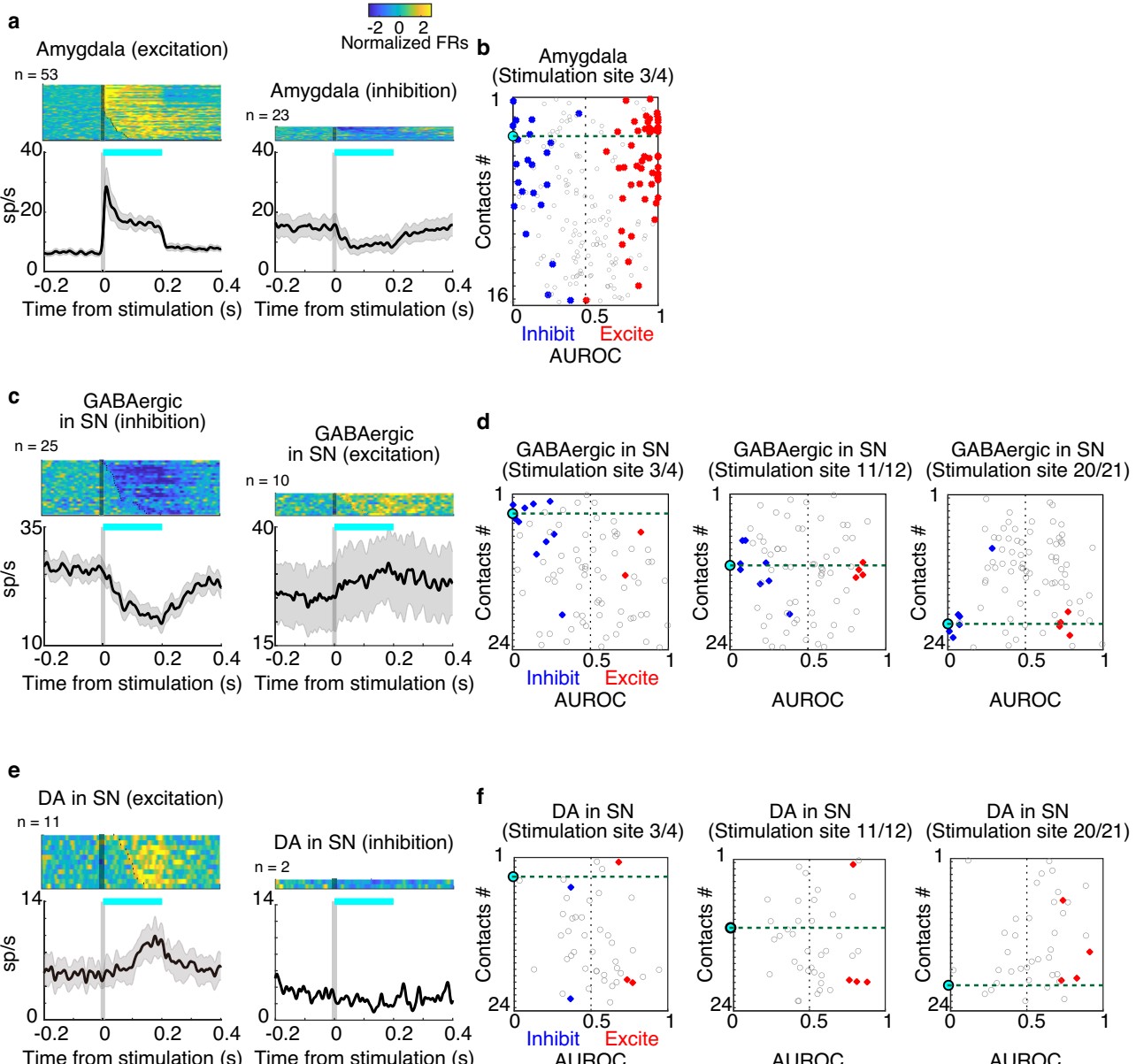

**Fig. 3 | Neuronal modulation by optogenetic stimulation. a** Population activity of amygdala neurons that were excited (left) or inhibited (right) by optogenetic stimulation. Spike activity was smoothed with a Gaussian kernel ($\sigma = 10$ ms). The shaded gray area indicates ±1 SEM for the averaged population activity. Responses of individual neurons (upper) were converted to the color scale and sorted by modulation latencies (dots). The duration of optical stimulation was 200 ms (cyan bar). **b** Data from combined all sessions recording amygdala neurons while optogenetically stimulating the amygdala. The degree of the modulation by the stimulation is defined as the area under the receiver operating characteristic curve (AUROC) based on the activity in stimulation trials vs. control (sham stimulation) trials for each neuron. Each dot indicates the AUROC score in each neuron. The red

or blue-filled marker shows statistically significant changes of the neuronal responses by optogenetic stimulation ($P < 0.05$, two-sided paired $t$-test). AUROC > 0.5 or <0.5 indicates that the neuron was excited or inhibited by stimulation comparing with non-stimulation trials, respectively. **c, e** Population activity of putative GABAergic or putative DA neurons that were inhibited or excited by optogenetic stimulation of the amygdala's axon in substantia nigra. The shaded gray area indicates ±1 SEM for the averaged population activity. **d, f** Data from combined all sessions recording putative GABAergic or putative DA neurons while stimulating the amygdala's axon in the substantia nigra by one of the stimulation ports of fiber optic (left: 3rd/4th, center: 11th/12th, right: 20th/21st). The same formats of **d**.

putative GABAergic neurons (Fig. 4e), but not for putative DA neurons (Fig. 4f). Among optically activated putative GABAergic neurons (Fig. 4e), cells that were close to the O-stim (within 0.25 mm; $n = 17/25$, 68 %) were inhibited quickly (mean: 47.6 + − 31.9 ms). The remaining neurons beyond 0.25 mm ($n = 8/25$, 32 %) were inhibited significantly, but more slowly (mean: 93 + − 51.3 ms, $t$-test $P = 0.0453$).

These data suggest the direct connections of amygdala neurons to GABAergic neurons in SN, as shown in Fig. 4c. In this schematic diagram, most amygdala neurons (shown by black circles, e.g., $n = 4$) are infected with viral vector (blue ovals) and express ChR2 in their axons (blue

lines). Then, each GABAergic neuron would receive inputs from some of the amygdala neurons. These inputs (i.e., axons) are close with each other to make synapses to the GABAergic neuron. In our experiments, O-stim was localized ($n = 3$) in SN (Fig. 3d, f). If one GABAergic neuron is close to the O-stim position (Fig. 4c, #4, red circle), it is highly affected by the inputs from amygdala by the multiple axons activated by O-stim (Fig. 3d). The input from the amygdala is mostly inhibitory postsynaptic potential (IPSP) (Fig. 3c, d). Instead, GABAergic neurons away from O-stim were less affected by O-stim (Fig. 3d) with longer latency (Fig. 4e) and less inhibition (Figs. 3d and 4h). This is predicted by Fig. 4c.

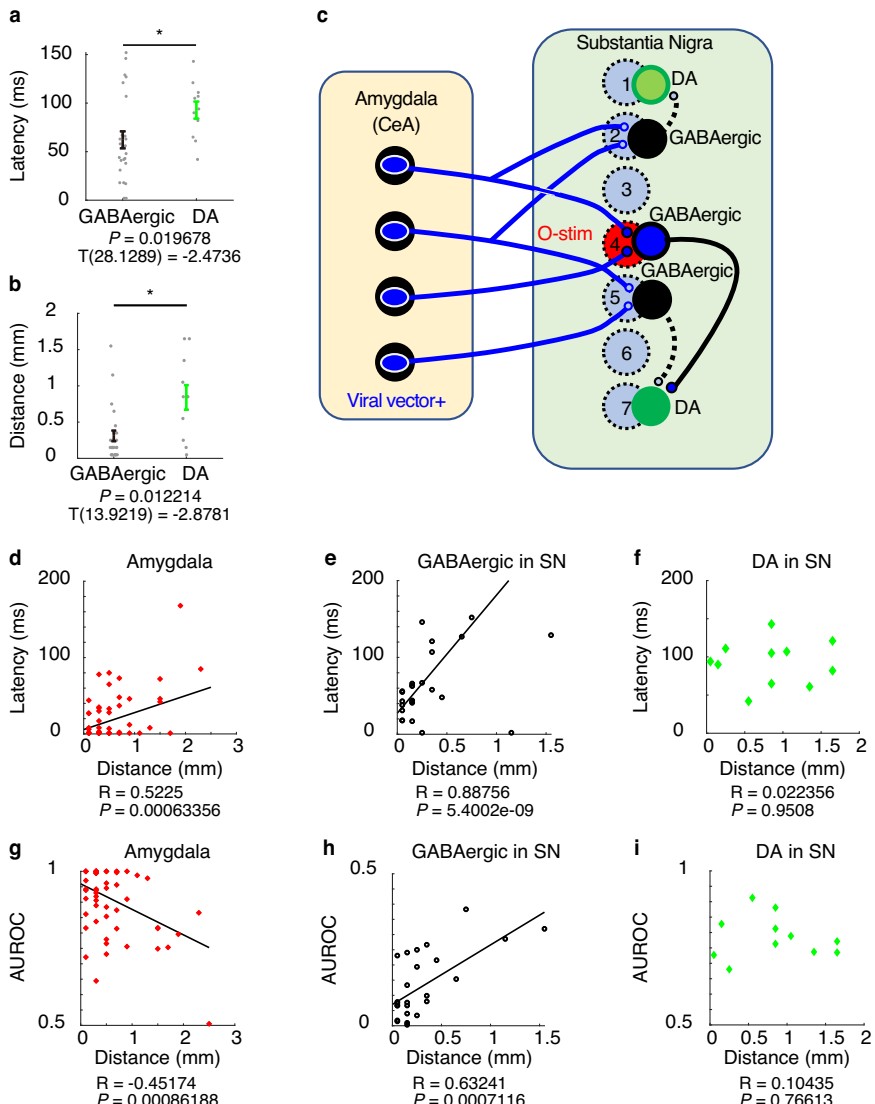

**Fig. 4 | Amygdala-GABAergic neuron in SN-DA neuron circuit and environmental information. a** Averaged latency for the neuronal modulation started by optogenetic stimulation. Error bars show ±1 SEM, and each gray dot represents one neuron. Responses are inhibited for putative GABAergic neurons ($n = 25$) and excited for putative DA neurons ($n = 11$). **b** Distance between effective optical stimulation and recording sites for putative GABAergic neurons, ($n = 25$) and putative DA neurons ($n = 11$). Error bars show ±1 SEM, and each gray dot represents one neuron. A two-sided paired $t$-test was applied for statistical testing (**a**, **b**).

**c** Schematic diagram showing the effect of optogenetic stimulation on the amygdala-SN circuit. **d–f** Relationship between distance (abscissa) and latency (ordinate) in each area. Excited neurons in the amygdala ($n = 53$), inhibited putative GABAergic neurons ($n = 25$), and excited putative DA neurons ($n = 11$) are shown in **d**, **e**, and **f**, respectively. **g–i** Relationship between distance (abscissa) AUROC (ordinate) in each area. AUROC close to 1 means stronger excitation, while AUROC close to 0 means stronger inhibition (compared with AUROC 0.5). Same population as **d–f**. The $P$-values were not corrected for multiple comparisons (**d–i**).

In addition, some putative DA neurons were also modulated by O-stim (Fig. 3e). However, neither the latency (Fig. 4f) nor the magnitude (Fig. 4i) of their responses showed any relation to the distance between O-stim and the recording site. These data suggest that DA neurons rarely receive direct inputs from the amygdala. Instead, some GABAergic neurons in SN, which receive direct inputs from the amygdala (Fig. 4c, #4), could modulate DA neurons, which may be located away from the GABAergic neurons (e.g., Fig. 4c, #7), by sending their axons to them. Accordingly, the O-stim would affect DA neurons that are located at random positions depending on the GABAergic SN neuron-DA neuron circuit (as illustrated in Fig. 4c).

Since O-stim is localized in each experiment (as shown in Fig. 4c, red), GABAergic neuron close to the stimulation may get multiple synaptic inputs from multiple amygdala neurons (Fig. 3d), which would be effective (mainly inhibitory, Fig. 3c). In contrast, DA neuron

may receive inputs from multiple GABAergic neurons, but only a single GABAergic neuron may receive the stimulation effect (Fig. 4c, #7), which may not be effective. Moreover, another DA neuron may receive inputs from GABAergic neuron(s) which do not receive the stimulation effect (Fig. 4c, #1). Overall, the effect of O-stim on DA neurons is not so strong as compared to GABAergic neurons.

Taken together, these data shed light on the organization of the amygdala-GABAergic SN neuron-DA neuron circuit. Our results indicate that both the amygdala-GABAergic SN neuron and the GABAergic SN neuron-DA neuron connections are inhibitory. Thus, the main influence of the amygdala on DA neurons is excitatory and is mediated by disinhibition through GABAergic neurons in SN (Fig. 2c, e, g; left side). In fact, anatomical studies have shown that the main projection from the amygdala to SNr is inhibitory and originates from CeA[8,25,26]. Likewise, neurons in SNr are well known to be inhibitory[27–30] and project to DA neurons[3,24], in addition to SC neurons[31].

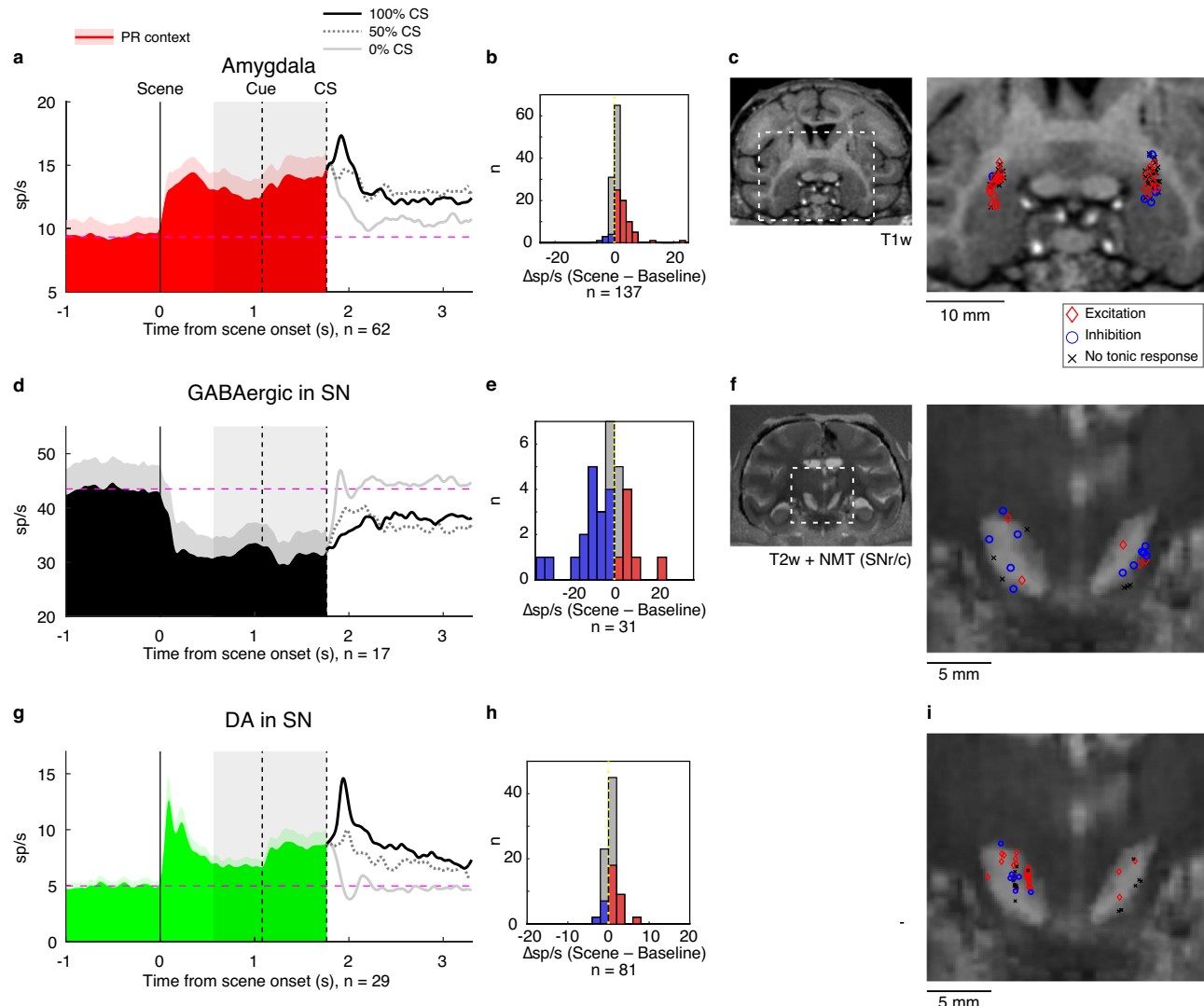

**Fig. 5 | Tonic modulation in environmental contexts. a, d, g** Population activity for amygdala (excited responses), putative GABAergic (inhibited responses), or putative DA (excited responses) neurons in the Pavlovian task (PR environment). The shaded gray area indicates the spike-count window used to calculate average activities for the tonic component of the scene responses (0–1.2 s before the CS onset). Tonic excitation was shown in many of amygdala and putative DA neurons. Tonic inhibition was shown in many of putative GABAergic neurons. The shaded red area indicates +1 SEM for the averaged population activity. **b, e, h** Distribution of responses to PR scene for amygdala, putative GABAergic, and putative DA neurons of all types (excited, inhibited, and non significant). The difference in firing rates was calculated by subtracting the baseline response (500 ms before scene onset) from the tonic response in each neuron. **c** Recording locations of amygdala neurons in Monkey SO superimposed on MRI image. The dotted region shows the location of magnified detail. Recording sites showing excited and inhibited responses to PR scene are indicated by red diamond and blue circle markers, respectively ($P < 0.05$, two-sided paired $t$-test). Cells with no significant response to PR are indicated by black cross markers. **f, i** Recording locations of putative GABAergic or putative DA neurons in Monkey SO. The same format as in **c**.

## Responses of amygdala, GABAergic neurons, and DA neurons to environmental contexts

Thus far, we have seen that putative DA neurons are sensitive to environmental contexts in addition to objects (Fig. 1). Optogenetic probing of functional circuitry then raised the possibility that the amygdala provides environmental information to DA neurons (Figs. 2 and 3). In light of this hypothesis, we proceed here to evaluate the impact of environmental contexts on the amygdala, GABAergic neurons, and DA neurons in the Pavlovian task.

Many neurons showed tonic firing rate changes following the onset of PR-environment, to varying degrees across the three areas (Fig. 5, S4). The amygdala neurons displayed tonic excitation ($n = 62$, Fig. 5a) more often than tonic inhibition ($n = 8$, Fig. S4a-left, also Fig. 5b). These neurons were recorded mostly in the dorsal part in the region near CeA (Fig. 5c-right). The putative GABAergic neurons exhibited tonic inhibition ($n = 17$, Fig. 5d) more often than tonic

excitation ($n = 7$, Fig. S4b-left, also Fig. 5e), and vice versa for the putative DA neurons (tonic excitation: $n = 29$, Fig. 5g; tonic inhibition: $n = 9$, Fig. S4c-left, also Fig. 5h). The putative DA neurons with tonic excitation were found relatively anterior and dorsal part of the nigra (Fig. 5i and Fig. S5). The recorded putative GABAergic neurons were found in the dorsal part as well as in the ventral part of SN (Fig. 5f). The dorsal part may be included in SNc that is primarily composed of DA neurons, but contains a small number of GABAergic neurons[32,33]. In this study, we analyzed those as a group of putative GABAergic neurons.

Then, after CS appeared, the tonic activity continued because the scene image (e.g., PR-environment) remained until the end of the trial. This suggests that the tonic activity is based on PR-environment, not CS. In addition, these neurons (amygdala neurons, putative GABAergic SN neurons, putative DA neurons) changed their activities phasically after CS appeared (amygdala: 100% > 50% > 0%, GABAergic: 0% > 50% > 100%, DA: 100% > 50% > 0%), These data suggest that

amygdala–SNr GABAergic neuron–SC circuit is controlled by PR-environment with tonic activity and by reward-prediction by CS. However, the tonic activity almost disappeared when CS indicated 0% reward (as shown in Fig. 5a, d, g).

The overall results suggest that amygdala neurons may relay information about the context to the nigra, which in turn may cause these downstream neurons to distinguish between PR/NR scenes and associated CS fractals. To test this idea explicitly, we compared neuronal activities with the effects of O-stim which were recorded during the Pavlovian task. In two out of three cases, putative GABAergic neurons that were tonically inhibited by the environment were also inhibited by the O-stim. Likewise, three out of eight putative DA neurons tested had congruent excitatory responses both during the task and to the O-stim (Fig. S6). These data further support the interpretation that the information about environmental contexts is sent from the amygdala to DA neurons through disinhibition via GABAergic neurons in SN.

Notably, the tonic firing rate changes induced by PR environments in all three areas (both excitatory and inhibitory responses) persisted as long as the possibility of reward remained. However, this persistent activity was truncated by the onset of the 0% CS that foreclosed the possibility of reward on the current trial (Fig. 5). This finding indicates that tonic responses to the environment were not simple visual responses, but were rather associated with motivation (i.e., expectation of reward outcome).

We further considered whether such different neuronal responses to scene images could be attributed to visual stimulus features rather than value information. To test this notion, the Pavlovian task was repeated with a second set of images for the PR condition, and the pattern of responses was compared across the two sets (Fig. S7a). The responses to the two PR-environments were significantly correlated in these sets for the amygdala, putative GABAergic, and putative DA neurons ($P < 0.05$, Pearson's correlation, Fig. S7b–d). Our results confirm that these neurons were indeed sensitive to environmental information, rather than the visual features of individual scene images. These data further indicate that a subset of DA neurons mainly showed tonic excitation in the appetitive PR-environment like amygdala neurons, forming a striking contrast to tonic inhibition of GABAergic neurons, which is in harmony with the pattern of responses to the optogenetic stimulation (Figs. 3 and 4).

## Discussion

We used optogenetics and multi-channel electrophysiological recording to reveal that amygdala neurons, GABAergic neurons in SN, and DA neurons are sensitive to environmental context. Our data suggested that tonic activity changes in DA neurons are generated by the amygdala-derived indirect input through GABAergic neurons in SN. These data suggest that the amygdala regulates DA cell activity in order to maintain motivational conditions in particular environments.

Previously, it has been shown that BG control choices of objects for the sake of reaching a desired outcome (e.g., reward). This function is controlled by two sets of parallel BG circuits, namely the CDh circuit for flexible value and short-term memory, and the CDt circuit for stable value and long-term memory[34]. For both circuits, the direct pathway mediates the choice of good objects, whereas the indirect pathway mediates the rejection of bad objects[35]. These parallel pathways are necessary to make appropriate decisions, for instance, by taking advantage of the past experience with objects when possible. However, the goals of behavior can be contingent on diverse information in addition to a particular targeted object, such as the animal's current environment. Optimal behavior requires that brain areas encoding such higher-order factors connect to decision-making areas that integrate information from multiple sources and ultimately guide behavior. Our results suggest that the amygdala encodes environmental information and sends it to the BG circuits to modulate GABAergic and DA neuron responses in SN. Consistent with this idea, previous studies suggest that the amygdala may encode environmental information related to emotion (e.g., wonderful, worried, etc.[36]).

Animals often choose or reject particular objects to achieve favorable outcomes (e.g., reward). Previously the mechanisms of such goal-directed behavior have mostly been studied using very simple laboratory tasks. However, good or bad objects are often not easy to find in real life and may be located only in particular environments. For example, many animals respond to changes in food availability by seeking out new environments[37,38]. While favorable environments do not guarantee favorable outcomes, they often increase the probability of findings sought after objects or resources of interest. Accordingly, entering a noteworthy environment can prime specific goal-directed behaviors and neural pathways even before animals encounter objects for the goal.

In this study, we used large visual scene images to represent environments that predict different emotional outcomes (e.g., reward). These scene images can be distinguished from fractal CSs which have been utilized to find dopamine neurons in the primate SNc in many previous works[21,22]. The fractal CSs are primarily small and may also subserve as saccadic target cues. In our experimental task, a large scene image first appears as environmental information before appearing the fractal CSs that are directly associated with reward or punishment. Such information changes neuronal activities and animal behaviors, because the animals may prepare for subsequent events related to the CSs while the scene image appears. In fact, the behavioral changes based on the environmental information are different from those according to target cues such as fractal CSs, as demonstrated in our prior study[6]. In this paper, we clearly discriminated the proportions of reaction time changes depending on the environmental information and target information, and considered that this fits the Linear Approach to Threshold with Ergodic Rate model[39].

Could the persistent tonic activity change we observed be attributed to the persistence of one large visual image for the duration of each trial? Two findings argue against this interpretation. First, tonic activity was halted by the onset of the 0% CS fractal that foreclosed the possibility of reward, despite the continued presence of the large background image (Fig. 5, S4). Second, the same tonic activity changes were induced by two PR images with very distinct visual features (Fig. S7). Thus, GABAergic and DA neurons in SN only responded to environmental stimulus images insofar as they predicted likely outcomes.

One of the critical differences between tonic and phasic activity changes is how such responses continue during the period of an event. Since the tonic activity changes occur in a linear, ramp-like manner throughout the event, it is important to sustain the information about the current status until it changes. On the other hand, even if the phasic activity does not maintain response changes throughout the event, it seems critical to evaluate each step of a sequential action as individual steps may not be maintained consistently. From this viewpoint, it may not be plausible to discriminate simply the definitions of tonic and phasic activity changes just by the difference in the response duration. Yet, the tonic activity should maintain response changes during the period of the event.

In most previous studies, electrophysiological investigations on DA neuron activity focused on its phasic components, for instance, responses to reward outcomes or reward-predicting visual stimuli[21,22]. Here we have found that DA neurons can also respond to stimuli with tonic activity changes (see Figs. 1 and 5). Likewise, studies in rodents reported the tonic modulation of DA concentration measured by microdialysis or in vivo voltammetry in various environments[40–42]. In these experiments, rodents locomoted through spatially expansive environments, in a manner analogous to the large visual stimuli used as environments in the current study. This suggests that entering a new environment can induce tonic changes in DA neurons in many animals, including at least both monkeys and rodents. Other studies in monkeys

also demonstrated the tonic modulation of DA neuron responses which occurred under uncertain value conditions or during continuous changes of expected reward outcome[43–46].

Environmental and various underlying contexts (e.g., uncertainty or continual change) are important in real life for all animals[47,48]. Moreover, environmental information is related to the therapeutic effectiveness for some diseases, including Parkinson's disease[49,50]. It is suggested that animals exposed to enriched environments exhibit resistance to Parkinsonian symptoms[51]. Furthermore, environments affect the sensitivity to some drugs that are believed to target midbrain DA neurons[52]. Our findings suggest that environmental information could promote neuronal activation, signaling, and plasticity through DA circuits, with important consequences for human and animal wellbeing.

For each amygdala neuron, ChR2 is often located inside its cell soma, so that O-stim can activate it directly or inhibit it indirectly, as shown in Figs. 3b and 4c. For the GABAergic neuron, however, ChR2 is not located inside its cell soma but within the axon terminal of the amygdala neuron. Moreover, O-stim can modulate the GABAergic neuron activity if the axon terminal of the amygdala neurons has a synaptic connection to the GABAergic neuron. Then, the effect of O-stim is exerted on a single GABAergic neuron that is localized around the axon terminal, as shown in Figs. 3d and 4c. These data suggest that GABAergic neurons receive direct inputs from amygdala neurons.

Unlike the GABAergic neurons, the O-stim effect on DA neurons was not spatially selective, less common (Fig. 3f), and more delayed (Fig. 3e). These data suggest that DA neurons mostly do not receive a direct connection from amygdala neurons (unlike the GABAergic neurons). This indicates that the input from the amygdala is based on an indirect connection. Anatomical and physiological studies so far suggest that there are two groups of neurons in SN: SNr neurons (GABAergic) and DA neurons[3,24]. Since O-stim was localized within SN, the indirect connection is likely to be mediated by the GABAergic neurons.

According to the indirect connection to DA neurons, the effect of O-stim would be less clear, according to the image shown in Fig. 4c. Since O-stim is localized in each experiment (as shown in Fig. 4c, O-stim), GABAergic neurons close to O-stim may get multiple synaptic inputs from multiple amygdala neurons, which would be effective (mainly inhibitory, Fig. 3c). In contrast, DA neurons may receive inputs from multiple GABAergic neurons, but only a single GABAergic neuron may receive the O-stim effect (Fig. 4c, bottom), which may not be effective. Moreover, another DA neuron may receive inputs from GABAergic neuron(s), which do not receive the O-stim effect (Fig. 4c, top). Overall, the effect of O-stim is not so strong as compared to GABAergic neurons.

These data may suggest that our data analysis is less clear. But we propose that this method would raise new data and suggestions about the neuronal circuit mechanisms (amygdala–GABAergic neuron in SN–DA neuron), not only one particular brain area. For example, we propose that the amygdala operates on two distinct circuits within BG, with functional consequences manifest on both an immediate and a long-term time scale. Behavior executed over the course of a single trial is achieved on the immediate time scale and mediated by the pathway representing Action in the Environment in Fig. 6. We previously showed that amygdala neurons respond to emotional environments tonically and control eye movements via a direct inhibitory pathway to GABAergic neurons in SN[6,12]. Saccades are triggered through disinhibition when the tonic inhibitory influence of SNr on SC is paused[30]. However, this mechanism alone is insufficient to account for complex behavior, such as learning from the past experience, or for taking uncertainty or environmental contingencies into account.

Such higher-order influences on behavior depend on synaptic plasticity regulated by dopaminergic projections acting on CD, which

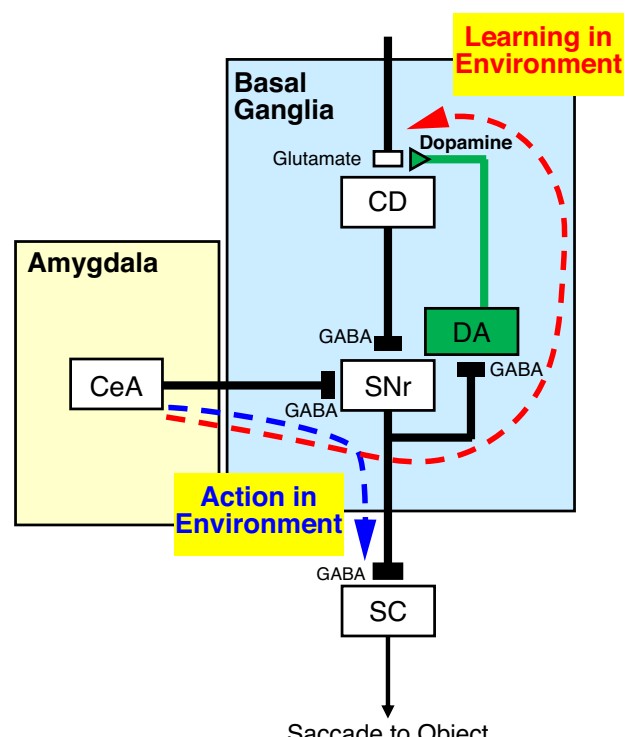

**Fig. 6 | Two pathways arising from amygdala for controlling BG circuit.** The solid lines indicate the anatomical pathways of GABAergic or Dopaminergic terminals. The dotted lines show hypothetical information flow through the amygdala-BG circuits for learning and action in environmental contexts.

constitutes the major input structure of BG (Learning in Environment in Fig. 6). To prove the idea, inhibiting SNr while recording the effect of amygdala stimulation on DA neurons is very important. However, there are at least two difficulties to perform this experiment. First, it is difficult to inhibit SNr neurons selectively because many DA neurons are often located among SNr neurons[53]. Second, the inhibition of SNr neurons causes a very strong behavioral effect: unexpected and repeated saccades (mostly to the contralateral side)[30]. In this study, unlike SNr neurons that were inhibited quickly and neurons close to the site of O-stim, DA neurons were excited slowly and distally from the stimulation site (Figs. 3 and 4). This suggests that amygdala neurons first inhibit GABAergic neurons in SN directly, and the circuit subsequently excites DA neurons. This disinhibition is due to the fact that all SNr neurons are GABAergic[30] and have local connections to DA neurons[3,24] in addition to their projections to SC (Fig. 6).

Most previous studies have shown that learning is caused by individual objects, primarily by the predicted outcomes[4,54,55]. Our present results indicate that the amygdala may provide an additional source of learning to the BG circuits, namely contingencies imposed by the environment.

## Methods

### Data reporting

No statistical methods were used to predetermine sample size. The experiments were not randomized, and the investigators were not blinded to allocation during experiments and outcome assessment.

### Animals and surgical procedures

We used two rhesus monkeys (*Macaca mulatta*) (monkey SO: 8.5 kg, 9 y old, male, monkey BA: 9.0 kg, 8 y old, male). All animal care and experimental procedures were approved by the National Eye Institute Animal Care and Use Committee (proposal number: NEI-622) and

complied with the Public Health Service Policy on the Humane Care and Use of Laboratory Animals. Both animals underwent surgery under general anesthesia during which a head holder and a recording chamber were implanted on the head. Based on a stereotaxic atlas (Saleem and Logothetis, 2007), we implanted a rectangular chamber targeting the amygdala and SN. The chamber was tilted anteriorly by 15 degrees in both monkeys. After confirming the position of the recording chamber using MRI (4.7 T, Bruker), a craniotomy was performed during a second surgery, and the time interval between those surgeries was >2 months. Anesthesia for those surgeries was induced using intramuscular injection of ketamine (5–15 mg/kg) and dexmedetomidine (0.002–0.032 mg/kg). After the placement of an endotracheal tube and IV catheter, the endotracheal tube was attached to the ventilator for the administration of isoflurane gas anesthesia (1–3%, to effect) and was maintained on gas throughout the procedure. Appropriate antibiotics were administered -24 h before surgery and after surgery continued for a total of 10 days. Butorphanol (0.05–0.15 mg/kg) or Ketoprofen (2–4 mg/kg) was administered after surgery as used for analgesics. Animals were placed on water control to perform experimental tasks appropriately. The fluid control was conducted in full accordance with the guidelines by the NIH Animal Research Advisory Committee. Briefly, the institution of water control was gradual in order to allow physiologic adaptation. Water may be withheld for no more than 24 h. First, the total amount of water provided daily was reduced at a rapid rate from full ad-lib water access down to the level of 400 ml daily. Then, the total amount of water was reduced at a rate of 50 ml every 24–48 h to the level of 300 ml daily. After this, the water was reduced at a maximum rate of 25 ml daily, until the monkey begins to perform the tasks readily for the water. In this study, the total fluid intake did not drop below 200 ml per day for five or more days. Routinely animals have been placed in the controlled fluid paradigm and the well-being of the animals has been carefully monitored by investigators and facility staff.

## Electrophysiological recordings

In each session one or two multi-site (16, 24, or 32 contacts) linear electrodes (V-probe or S-probe, Plexon) were lowered into the brain using an oil-driven micromanipulator system (MO-97A, Narishige). The micromanipulators were moved independently into the amygdala and/or substantia nigra while identifying electrophysiological indicators of gray and white matter boundaries. We allowed 60 min for the electrodes to stabilize before starting data acquisition and the behavioral protocol. Signals were pre-amplified and stored at 40 kHz for offline processing (OmniPlex, Plexon). In real time, signals were band-pass filtered between 0.2–10 kHz, and online spike sorting was performed using custom software implementing a voltage and time window discriminator (Blip). Analysis was based on offline spike sorting using the Kilosort algorithm followed by manual curation in the Phy[56].

## Pavlovian procedure

After a recovery period of >6 weeks after the above implantation surgery, each monkey was trained to become accustomed to the head restraint and experimental apparatus. Next, each monkey was trained in the following Pavlovian procedure before and after the craniotomy surgery. Each trial began with the appearance of a scene image (50 degrees diameter) that signaled either PR (possible reward) or NR (no reward) environment and remained on present for the duration of the trial (Fig. 1). In the PR environment, one of three conditioned stimuli (fractal objects) could appear that were associated with a liquid reward (water) as an unconditioned stimulus with either 100%, 50%, or 0% probability. Water was delivered through a sipper tube positioned in front of the monkey's mouth. In the NR environment no reward was forthcoming, and the unconditioned stimulus was an air-puff (20–30 psi) delivered through a narrow tube placed 6–7 cm from the face. Three different fractal objects were used as conditioned stimuli

on NR trials, and were associated with air-puff probabilities of 100%, 50%, and 0%. After 1.08 s of the scene presentation, a timing cue (white square, 2 degrees) appeared in the center of the screen. After 0.68 s, one CS appeared and remained present for 1.5 s, at which point offset of CS and delivery of corresponding US occurred simultaneously. The monkeys were not required to look at these images at any point in the trial. After presenting the scene image 1.08 s, the timing cue was presented. Thirty-two trials of each condition (100%, 50%, and 0% CS, crossed with PR and NR environments) were presented in pseudorandom order, for a total of 192 trials.

## Experimental control

All behavioral tasks were controlled by a custom system for neuronal recording and behavioral control system (Blip; available at http://www.robilis.com/blip/). The monkey sat in a primate chair facing a front-parallel screen in a sound-attenuated and electrically shielded room. Visual stimuli were rear-projected onto a screen by a digital light processing projector (PJD8353s, ViewSonic). Eye position was sampled at 1 kHz using a video-based eye tracker (EyeLink 1000 Plus, SR Research).

## Viral injections and optogenetics

We injected an adeno-associated virus type 2 vector (AAV2-CMV-ChR2-EYFP: $9.0 \times 10^{12}$ genome copy/ml) into the amygdala of one hemisphere in both monkeys (monkey SO: left amygdala, monkey BA: right amygdala). Two penetrations in monkey BA and three penetrations in monkey SO were made into one side of the amygdala at least 1.41 mm apart from each other. For each penetration, 2 μl (for monkey BA) and 2 or 3 μl (for monkey SO) of the vector were introduced at a rate of 0.4 μl/min for the first 0.2 μl, followed by 0.08 μl/min for the remainder of the injection controlled by a 10 μl Hamilton syringe and motorized infusion pump (Harvard Apparatus, Holliston, MA, USA). The vector was successfully used in the macaque brain in a previous study[12,57], and the injection location was verified with histological procedures described previously[12]. For histological procedures, the subject was immobilized with ketamine (10 mg/kg) and diazepam (1 mg/kg), deeply anesthetized with an overdose of sodium pentobarbital (390 mg/ml) and then perfused transcardially with 0.1 M phosphate-buffered saline (PBS) followed by 4% paraformaldehyde in PBS at a pH of 7.4. The head was fixed to the stereotaxic frame, and the brain was cut into blocks in the coronal plane including midbrain region. The block was post-fixed overnight at 4 C°, and then cryoprotected for 1 week in increasing gradients of glycerol solution (10–20% glycerol in PBS) before being frozen. The frozen block was cut into 50 μm sections using a microtome.

To perform O-stim and electrophysiological recording at the same time, we used 16 or 24 contact multi-site linear electrodes with one or three combined optic fibers (S-probe, Plexon). The light source was a 473 nm DPSS blue light laser with a maximum power of 100 mW (Opto Engine LLC). We left the laser on continuously during the experiment and placed a mechanical shutter switch (Luminos Industries Ltd) in the light path to turn the laser on and off. We measured the light intensity at the tip of the optrode before penetration of the brain using an optical power meter (1916-C, Newport Corporation) coupled with an 818-SL/DB photodetector. The maximum light intensities were set as 0.15 mW for the amygdala and 0.5 mW for the projection sites in the substantia nigra. Stimulation and non-stimulation periods were pseudo-randomly interleaved 200 times during free viewing while various visual stimulation regimes were presented (including static pictures, a blank screen, and movies).

## Identification of DA and GABAergic neurons in SN and amygdala neurons

We searched for DA and GABAergic neurons in and around the substantia nigra. DA neurons were identified as having <10 sp/s baseline

activity (0–500 ms before the scene onset), phasic excitation to the scene image comparing with baseline activity, and excitation to 100% reward CS comparing with 50% or 0% reward CS. GABAergic neurons in SN were identified as having >15 sp/s baseline activity and phasic inhibition to the scene image comparing with baseline. To reach the amygdala area, the electrode was first advanced through GPe and/or striatum areas. After passing through a quiet white matter region, spikes of amygdala neurons (heterogeneous firing patterns, relatively consistent firing with no pause) grew larger. Amygdala neurons that show phasic excitation to the scene image onset within 0–300 ms were further analyzed in this study.

In this task procedure, the period of the scene presentation before CS onset was 1760 ms which is the timing from the scene onset to CS onset through a timing cue. To identify phasic visual responses to the scene image, the responses in an initial segment of the scene presentation, in which we used a 0–300 ms window after the scene onset for DA and amygdala neurons and 0–500 ms for GABAergic neurons, were compared with the baseline activity ($t$-test). Regarding the tonic component of the scene response, we calculated the responses in later and longer segments, which was a 0–1200 ms window before the CS onset, and compared them with the baseline activity ($t$-test). To calculate the responses to CS or US, 0–300 ms after those onsets were compared with the baseline activity ($t$-test).

### Neuronal activity analyses

For the analyses of neuronal activity, spike-density functions (SDFs) were generated by convolving spikes times with a Gaussian filter ($\sigma = 20$ ms). To investigate neuronal responses to the O-stim, we compared neuronal activities in a 200 ms window after stimulation vs. non-stimulation events for each neuron. Significance was assessed with an alpha threshold of 0.05. A modulation index was computed for each neuron, defined as the area under the receiver operating characteristic curve (AUROC) comparing responses to stimulation vs. non-stimulation.

The latency of responses to O-stim was measured by computing the earliest time that the neuron's activity exceeded (for excitation) or fell below (for inhibition) the baseline by two standard deviations for at least 7 out of 10 consecutive ms bins. Neurons that failed to meet this criterion were excluded from the analysis ($n = 9$).

### Statistical analyses

All data were preprocessed and analyzed using custom programs written in MATLAB. Two-sample $t$-test was used for the mean difference comparison for neural responses or analysis of stimulation effects (two-sided). Error-bars in all plots show the standard error of the mean (SEM). The significance threshold for all tests in this study was $P < 0.05$. ns: not significant, *$P < 0.05$.

### Reporting summary

Further information on research design is available in the Nature Portfolio Reporting Summary linked to this article.

## Data availability

Source data are provided in this manuscript as source_data.xlsx. The datasets generated and analyzed during this study are available on figshare at https://doi.org/10.6084/m9.figshare.19105136.v1. Source data are provided with this paper.

## Code availability

All pre-processing and analyses were performed using Matlab 2020b or 2022b. Code used for analysis and figure generation in this manuscript is available on figshare at https://doi.org/10.6084/m9.figshare.19105136.v1.

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

## Acknowledgements

We thank D. McMahon, M.K. Smith, D. Parker, G. Tansey, D. Yu, A.M. Nichols, D. Yochelson, J. Fuller-Deets, A.V. Hays, and M Fujiwara for technical assistance. This work was supported by Intramural Research Program at the National Eye Institute, the National Institutes of Health, United States (project number: 1ZIAEY000415, O.H., https://reporter.nih.gov/project-details/9796699), AMED Grants (JP20dm0207077 and JP21dm0207077, M.T.), and MEXT/JSPS KAKENHI Grant (JP19H05467, M.T.).

## Author contributions

K.M. and O.H. designed this research, performed the experiments, analyzed the data, and wrote the manuscript. K.I. and M.T. produced the viral vector. K.M., K.I., M.T., and O.H. discussed the results and reviewed and edited the manuscript.

## Competing interests

The authors declare no competing interests.

## Additional information

Kazutaka Maeda or Okihide Hikosaka.

