## [Peer Review File · Nature Communications]

Environmental context-dependent activation of dopamine neurons via putative amygdala-nigra pathway in macaquesREVIEWER COMMENTS

Reviewer #1 (Remarks to the Author):

The authors used simultaneous optogenetic excitation and multi-electrode recording in 2 awake monkeys to investigate amygdala (CeN) and midbrain (SNc vs SNr) activity during shifts in predictive environments on Pavlovian responding to positive and negative cues. They first confirm that in trained animals that DA phasic response followed by tonic excitation to positive environment cue; and that there is no tonic response to the 'no reward (air puff) environment. DA neurons were physically responsive to CS and US. Together these data showed 2 responses of DA neurons: tonic to environment and phasic to 'object'. Moving to optogenetic excitation of presumptive GABA projections from the CeN, they record and manipulate light in the midbrain, showing that excitation inhibits SNr (putative GABAergic neurons), and after slight latency, activates DA neurons. Following up with careful measurements and analyses, they show that neither magnitude or latency of DA responses had any relation to distance between optical stimulation of afferent fibers or recording site. Finally (fig 5) they demonstrate that a positive environmental context induces tonic excitation of the CeN, inhibition of putative SNr, and excitation of DA neurons. Recordings/measures during optogenetic stimulation, outside of the task, are convincing for identifying cell types (Figs 3,4). Overall, this is a technically challenging, carefully done study that looks at an important question-- how can higher levels of motivationally relevant (and temporally slower) sensory information influence immediate actions?

I have only one major comment. The idea that all non-DA neurons are in the SNr seems to be much too 'binary' and interpretation of how the SNc/r is put together, particularly in the monkey. 30% of histochemically defined neurons in the SNc are GABAergic. Therefore, lumping them into the SNr seems to do these data somewhat of a disservice. The CeN project broadly over the dorsolateral SNc and A8, and although there is some encroachment on the SNr, the finding of 2 paths (action and learning in environment, Fig 6) might imply different roles for GABAergic neurons in SNc versus SNr. (Fudge et al 2017)

Minor comments:

On page 6, section describing Fig 5, there is implication that the NR context as well as the PR context is evaluated for tonic firing rate change. But only the PR environment is described. I assume that this is due to floor effects/nonresponsivity for the NR condition, but this might be mentioned. In last paragraph (line 207), we return to controls for visual stimulus features for both PR and NR contexts, which leads one to think they have missed part of the results.

2. It would be great to see AP levels for ventral midbrain recordings/stimulations.

Reviewer #2 (Remarks to the Author):

In this manuscript, Maeda and colleagues employed optogenetics and single-unit recordings in two monkeys to examine how the central amygdala might modify dopamine neuron activity to reflect environmental contingencies. Their evidence is suggestive of a two-stage circuit, in which the central amygdala inhibits GABAergic cells of the substantia nigra pars reticulata, which then inhibit substantia nigra dopamine (DA) neurons.

Understanding the circuit mechanisms that allow DA neurons to reflect the environment is a laudable goal. However, I have numerous concerns about the methods and interpretation that significantly dampen my enthusiasm for the manuscript.

Major concerns:

- The authors make major interpretative leaps that are not supported by the data:
 - o The authors' first conclusion, which is in the title of the manuscript, is that DA neurons are "activated in rewarding environment by tonic disinhibition from amygdala." This claim is not supported by the data. To make this claim, the authors needed to manipulate the amygdala and record DA responses during the reward context task. More precisely, the best experiment would be to inhibit the amygdala during context presentation and show that DA neurons no longer encode the context information. Instead, the authors manipulated amygdala activity separately, without any task; they used stimulation parameters that did not mimic the recorded data (e.g., the stimulation is an order of magnitude shorter in duration than the normal amygdala response to the context cue); and they did not attempt any inhibition. From their data, the most the authors can conclude is that some putative dopamine neurons are excited by stimulation of the amygdala; it is unclear if this relates to the encoding of contextual information. Indeed, only 3 of the 8 DA neurons that the authors recorded in both the optical stimulation task and the Pavlovian task had the same excitatory responses in both contexts (lines 196-197). Such findings do not make me confident that the amygdala plays a crucial role in providing DA neurons with contextual information, as the authors claim.
 - o The authors' second major conclusion is that the amygdala acts on DA neurons through SNr. (Indeed, in the discussion, they say "we have found that tonic activity changes in DA neurons are generated by the amygdala-derived indirect input through SNr"). But there's no direct evidence of this. Any other multi-synaptic path from amygdala to DA neurons could explain their findings. To make this conclusion, the authors would need to inhibit SNr while recording the effect of amygdala stimulation on DA neurons. They did not do this. At the very least, the authors need to tone down their conclusions. Their results may suggest that this disinhibitory process is possible (which we already knew from papers such as Steinberg et al., 2020, Neuron), but it certainly does not confirm that it happens.
- I am not convinced that the authors successfully identified DA neurons and separated them from non-DA neurons in the SNr. This calls into question all the rest of their findings. Figure S3 does not show a clear separation by baseline activity or spike duration. In addition, the authors appear to have required putative DA neurons to show phasic excitation to context or CS, and putative SNr neurons to show phasic inhibition to context. This is circular reasoning—the authors cannot identify these neurons based on their excitation or inhibition patterns, and then make independent claims about what these neurons do in the task. It seems quite possible that many of the neurons the authors claim are DA neurons are actually not DA neurons, which might explain the variability of recorded responses (with some neurons having longer-lasting responses than others). The authors should attempt to use other, more stringent, and non-circular methods to identify neurons, perhaps based on multiple features of the spike shape. Ultimately they may not be able to conclusively identify what neurons they are recording based on electrophysiology alone (this is a longstanding

problem in the field; see, for example, Margolis ... Fields, J Physiology 2006). At the very least, the authors should use “putative DA” rather than “DA,” to make it clear that they have not conclusively identified these DA neurons. And perhaps they can look more closely at those neurons with baseline activity or spike duration that is on the border between SNr and DA. Do these neurons have particular patterns of task activity?

- The authors make a tidy distinction between “phasic” and “tonic” dopamine activity and argue that phasic activity is modulated by the conditioned stimulus (CS) and tonic activity is modulated by the environmental context. But I do not understand how the authors distinguish between phasic and tonic, and how they distinguish between CS and environment. When I look at the neural activity, I see very little difference between the response to the environmental stimulus and the response to the CS (see, for example, the green response vs the black response in Fig 1a and Fig 5g). It seems to me that there are longer and shorter responses to both the environment stimulus and the CS. How do the authors define “phasic” vs “tonic”? Isn’t the environmental context just another conditioned stimulus?
- The authors can do a better job explaining what is novel and significant in this manuscript. In particular, what does this paper add to our understanding of the links between CeA and substantia nigra DA neurons, that isn’t already in Steinberg et al., 2020?

Minor comments:

- Why are the measured latencies, even in the amygdala neurons, so long? Previous work combining optogenetics and electrophysiology often report latencies less than 5 or 10ms; longer latencies were assumed to be polysynaptic.
- The sample sizes are unclear throughout. How does the example in Fig. 1 generalize to all other recorded neurons, and how many recorded neurons were there? For Figs. 3-5, please specify sample size for the statistics.
- Fig. 4c is overly speculative. Why is there a converging input from multiple CeA neurons to one SNr neuron? Another explanation for the results in e and h is that as the stimulation site is moved away from the recorded neuron, the light reaching the terminals will be dimmer and therefore less reliable in driving spikes. The data also does not directly support the projection from the recorded SNr neuron to the DA neurons. Finally, the title of Fig .4 is confusing. How does the data in this figure relate to ‘environmental information’?
- There is no data supporting the statement (in the abstract and last sentence of the manuscript) that ‘the amygdala provides additional source of learning to the BG circuit’. Did the neuronal response to the scene image develop over learning? Did optical stimulation of CeA affect learning?
- Fig. 5 shows the neural responses to the PR environment. How does the neural activity look like in the NR environment? Are SNr and amygdala neurons tuned to either PR or NR environment specifically, as the example DA neuron shown in Fig. 1?
- Figure 1: are trials in the green of the raster plots sorted in the same way as in the grey-black part? The sudden jump from one to three panels is a bit confusing.
- Line 84-85: ‘the absence in the environmental context’ is confusing. There is always a context for any form of task. Perhaps the authors mean in the absence of a context-predicting cue?
- Fig. S1c-d: It would be good to show the neural response to optical stimulation in a Chr2+ animal.
- Fig 3: typo in the y-axis of b, d, and f

Reviewer #3 (Remarks to the Author):

Maeda and colleagues sought to characterize the response of dopamine neurons to environmental/contextual cues and the source of such information in non-human primates. Monkeys were first presented with an aerial photograph (“scene”) which determines whether a given trial is associated with a potential reward (water) or air puff (possible-reward or no-reward; PR or NR, respectively) in a Pavlovian design. A scene stimulus was followed by a timing cue and a fractal image, which indicated the probability of water or air puff (conditioned stimulus, CS). On average, dopamine neurons showed a transient activation in response to the contextual cue, followed by a sustained elevation in firing, which was again elevated by a timing cue. CS’s then evoked transient activation or inactivation depending on the reward probability. The authors then show that optogenetic stimulation of amygdala neurons activates neurons in the substantia nigra pars reticulata (SNr) and inhibits dopamine neurons. The latency of response was shorter in SNr compared to dopamine neurons, suggesting activation of amygdala neurons disinhibits dopamine neurons via SNr. The authors then record the activity of amygdala and SNr neurons in the same task, and show that neurons in both areas exhibit tonic modulations in the period between the scene and outcome. The authors conclude that tonic activation of dopamine neurons by rewarding environmental cues depends on tonic disinhibition from amygdala.

This study addresses an important question and the authors performed relatively difficult experiments in non-human primates. The authors should be commended for their effort. However, there are some interpretational difficulties that limit the extent at which the authors can draw strong conclusions.

Major issues:

1. The authors use aerial photographs for contextual cues. There are two issues. First, it is very unlikely that monkeys can understand the meaning of the pictures – whether these pictures are “scenes” or representing an “environment”. It is most likely that these images are nonsense images to monkeys. Second, because of this reason, the difference between the contextual cues and conditioned stimuli (fractals) are mainly the size in the computer monitor. Therefore, I am not very convinced that the contextual cues had a special meaning that qualifies to be a “scene” or “environment” beyond providing broader task information in the task. Because of these reasons, whether the contextual cues and the CSs had fundamental differences that are processed by different neural circuits remains largely unclear. It remains unclear whether contextual cues and CSs of a same size can cause different firing patterns of the neurons studied in this work.
2. Although the tonic firing patterns are somewhat similar (or roughly mirror-image) between amygdala, SNr and dopamine neurons, their firing patterns have important differences. For example, dopamine neurons increased their tonic firing after the timing cue (Fig. 5g). Amygdala neurons have a weak but similar tendency, but SNr neurons did not change the tonic level after the timing cue. The pattern of phasic responses were also different. The authors use optogenetics to infer synaptic connectivity but have not performed manipulations during the task. Furthermore, it is unclear whether the amygdala neurons that the authors recorded indeed project to the SNr. Considering these limitations, the conclusion that disinhibition via SNr generates tonic modulations in dopamine neurons (which is in the title) appears to be premature.

3. The authors do not report response patterns of dopamine, SNr nor amygdala neurons. Were firing patterns of these neurons consistent with the author's model?

RESPONSE TO REVIEWERS

of “Dopamine neurons activated in rewarding environment by phasic and tonic disinhibition from amygdala”

by K. Maeda, K. Inoue, M. Takada, and O. Hikosaka

August 23, 2022

We would like to thank the reviewers for careful and thorough reading of this manuscript and for the thoughtful comments and constructive suggestions, which help to improve the quality of this manuscript. We agree with almost all their comments and we have revised our manuscript accordingly.

We respond below in detail to each of the reviewer’s comments. We hope that the reviewers will find our responses to their comments satisfactory, and we are willing to finish the revised version of the manuscript including any further suggestion that the reviewers may have.

Please, find below the reviewers’ comments repeated in black and our responses (blue) inserted after each comment. To facilitate the work of the reviewers, in some instances we refer to the newer manuscript indicating the page and we also submit the manuscripts that the changes are clearly highlighted.

Looking forward hearing from you soon.

Sincerely,

Kazutaka Maeda and Okihide Hikosaka

REVIEWER COMMENTS

Reviewer #1 (Remarks to the Author):

The authors used simultaneous optogenetic excitation and multi-electrode recording in 2 awake monkeys to investigate amygdala (CeN) and midbrain (SNc vs SNr) activity during shifts in predictive environments on Pavlovian responding to positive and negative cues. They first confirm that in trained animals that DA phasic response followed by tonic excitation to positive environment cue; and that there is no tonic response to the 'no reward (air puff) environment. DA neurons were physically responsive to CS and US. Together these data showed 2 responses of DA neurons: tonic to environment and phasic to 'object'. Moving to optogenetic excitation of presumptive GABA projections from the CeN, they record and manipulate light in the midbrain, showing that excitation inhibits SNr (putative GABAergic neurons), and after slight latency, activates DA neurons. Following up with careful measurements and analyses, they show that neither magnitude or latency of DA responses had any relation to distance between optical stimulation of afferent fibers or recording site. Finally (fig 5) they demonstrate that a positive environmental context induces tonic excitation of the CeN, inhibition of putative SNr, and excitation of DA neurons. Recordings/measures during optogenetic stimulation, outside of the task, are convincing for identifying cell types (Figs 3,4). Overall, this is a technically challenging, carefully done study that looks at an important question--how can higher levels of motivationally relevant (and temporally slower) sensory information influence immediate actions?

I have only one major comment. The idea that all non-DA neurons are in the SNr seems to be much too 'binary' and interpretation of how the SNc/r is put together, particularly in the monkey. 30% of histochemically defined neurons in the SNc are GABAergic. Therefore, lumping them into the SNr seems to do these data somewhat of a disservice. The CeN project broadly over the dorsolateral SNc and A8, and although there is some encroachment on the SNr, the finding of 2 paths (action and learning in environment, Fig 6) might imply different roles for GABAergic neurons in SNc versus SNr. (Fudge et al 2017)

Reply:

This suggestion about the different roles for GABAergic neurons in SNc versus SNr is very important. Similar to this view, DA neurons also have different anatomical and physiological properties from dorsal to ventral where TH-positive cells penetrate deep (Haber et al., J. Comp. Neurol, 1995, and Evans et al., Cell Rep 2020). But we could not clearly separate whether the high-firing rate neurons were only identified in SNr or not in this study. So then, to describe our findings without misinterpretation as reviewer's comments, we use the term "putative SNr neurons" which shows the high-firing rate patterns in substantia nigra area compared with "putative DA neurons" that show low firing rate patterns in some cases in the results section of the manuscript. In future studies, we would like to investigate the different functional roles of SN in the dorsal and ventral parts, including identifying DA and GABAergic.

References

- Haber, S. N., Ryoo, H., Cox, C. & Lu, W. Subsets of midbrain dopaminergic neurons in monkeys are distinguished by different levels of mRNA for the dopamine transporter:

comparison with the mRNA for the D2 receptor, tyrosine hydroxylase and calbindin immunoreactivity. *J. Comp. Neurol.* 362, 400–410 (1995).

- Evans, R. C. et al. Functional Dissection of Basal Ganglia Inhibitory Inputs onto Substantia Nigra Dopaminergic Neurons. *Cell Rep* 32, 108156 (2020).

Minor comments:

On page 6, section describing Fig 5, there is implication that the NR context as well as the PR context is evaluated for tonic firing rate change. But only the PR environment is described. I assume that this is due to floor effects/nonresponsivity for the NR condition, but this might be mentioned. In last paragraph (line 207), we return to controls for visual stimulus features for both PR and NR contexts, which leads one to think they have missed part of the results.

Reply:

We agree with your comment. To emphasize responses in PR context, we remove the word “NR” in this paragraph as shown below.

- Previous manuscript: “We further considered whether such different neuronal responses to PR and NR scenes could be attributed to visual stimulus features rather than value information.”
- Current manuscript (lines 214-215): “We further considered whether such different neuronal responses to scene images could be attributed to visual stimulus features rather than value information.”

Also, like what another reviewer commented, we generated a figure to show responses in NR contexts to help the review process. The data in NR environment are shown in “Fig. R1,” which is attached to the bottom of this document.

2. It would be great to see AP levels for ventral midbrain recordings/stimulations.

Reply:

We added a supplemental figure (Fig. S5) to show AP levels for the recording data.

Reviewer #2 (Remarks to the Author):

Major concerns:

- The authors make major interpretative leaps that are not supported by the data:
 - o The authors' first conclusion, which is in the title of the manuscript, is that DA neurons are "activated in rewarding environment by tonic disinhibition from amygdala." This claim is not supported by the data. To make this claim, the authors needed to manipulate the amygdala and record DA responses during the reward context task. More precisely, the best experiment would be to inhibit the amygdala during context presentation and show that DA neurons no longer encode the context information. Instead, the authors manipulated amygdala activity separately, without any task; they used stimulation parameters that did not mimic the recorded data (e.g., the stimulation is an order of magnitude shorter in duration than the normal amygdala response to the context cue); and they did not attempt any inhibition.

Reply:

We understand that your suggestion is very important: How do DA neuronal activity (and their functions) change if amygdala neurons are inhibited experimentally? In a previous study (Maeda et al., *iScience*, 2020), we actually inactivated amygdala neurons and found that saccades to the contralateral side were inhibited. This effect was based on another neuronal circuit: Amygdala – SNr – SC (superior colliculus) (which is shown in Fig. 6 in this paper).

We now would like to use this method (Inhibition of the amygdala) to study the effect on DA neurons (which we did not do before). However, there may be some issues with the effect of this method. First, since the inhibition of the amygdala modulates saccade, the monkey may not perform the goal-directed behavior properly (e.g., unable to choose a good object). Then, DA neurons would be controlled by various neuronal circuits that are sensitive to goal-directed behavior, including the caudate nucleus (Kim et al., *Cell*, 2015), lateral habenula (Hong et al., *J Neuroscience*, 2011), pedunculopontine tegmental nucleus (PPTg) (Hong and Hikosaka, *Neuroscience*, 2014), etc. which are controlled by various kinds of context. So, this method (Inhibition of the amygdala) may not examine the selective neuronal circuit (Amygdala – SNr – DA).

However, we find this method (Inhibition of the amygdala) very interesting because we may be able to study the functions and mechanisms of various neuronal circuits. In the near future, we want to examine this method, possibly together with multiple neuronal circuits in various contexts.

References

- Kim, H. F., Ghazizadeh, A. & Hikosaka, O. Dopamine Neurons Encoding Long-Term Memory of Object Value for Habitual Behavior. *Cell* 163, 1165–1175 (2015).
- Hong, S., Jhou, T. C., Smith, M., Saleem, K. S. & Hikosaka, O. Negative reward signals from the lateral habenula to dopamine neurons are mediated by rostromedial tegmental nucleus in primates. *J. Neurosci.* 31, 11457–11471 (2011).

- Hong, S. & Hikosaka, O. Pedunculo-pontine tegmental nucleus neurons provide reward, sensorimotor, and alerting signals to midbrain dopamine neurons. *Neuroscience* 282, 139–155 (2014).

From their data, the most the authors can conclude is that some putative dopamine neurons are excited by stimulation of the amygdala; it is unclear if this relates to the encoding of contextual information. Indeed, only 3 of the 8 DA neurons that the authors recorded in both the optical stimulation task and the Pavlovian task had the same excitatory responses in both contexts (lines 196-197). Such findings do not make me confident that the amygdala plays a crucial role in providing DA neurons with contextual information, as the authors claim.

Reply:

This is an important comment, and we now have been thinking about the difference in optical stimulation (O-stim) between the Amygdala, SNr, and DA neurons. As shown in Fig. 3, O-stim effect was common for Amygdala neurons but less common in SNr neurons and fewer in DA neurons. We now suggest that such differences are based on the relationship between the position of each neuron and AAV viral vector, which is shown in Fig 4c.

For each amygdala neuron, ChR2 is often located inside its cell soma so that O-stim can activate it directly or inhibit it indirectly, as shown in Figures 3b and 4c. However, for the SNr neuron, ChR2 is not located inside its cell soma but within the axon terminals of amygdala neurons. Moreover, O-stim can modulate the SNr neuron's activity if the amygdala neuron's axon has a synaptic connection to the SNr neuron. But then, the O-stim effect on 1 SNr neuron is primarily localized around it (i.e., amygdala axons), as shown in Fig 3d and 4c. These data suggest that SNr neurons receive direct inputs from Amygdala neurons.

Unlike SNr neurons, the O-stim effect on DA neurons was not spatially selective, less common (Fig 3f), and more delayed (Fig 3e). These data suggest that DA neurons mostly do not receive a direct connection from amygdala neurons (unlike SNr neurons). This indicates that the input from the amygdala is based on indirect connection. Anatomical and physiological studies so far suggest that there are two groups of neurons in SN: SNr neurons (GABAergic) and DA neurons (Tepper et al., 1995; Kim et al., 2015). Since O-stim was localized within substantia nigra, the indirect connection is likely to be mediated by SNr neurons.

According to the indirect connection to DA neurons, the effect of O-stim would be less clear, according to the image shown in Fig 4c. Since O-stim is localized in each experiment (as shown in Fig 4c, O-stim), SNr neurons close to O-stim may get multiple synaptic inputs from multiple amygdala neurons, which would be effective (mainly inhibitory, Fig 3c). In contrast, DA neurons may receive inputs from multiple SNr neurons, but only 1 SNr neuron may receive the O-stim effect (Fig 4c, bottom), which may not be effective. Moreover, another DA neuron may receive inputs from SNr neuron(s), which do not receive the O-stim effect (Fig 4c, top). Overall, the effect of O-stim is not so strong compared with SNr neurons.

These data may suggest that our data analysis is less clear. But we propose that this method would raise new data and suggestions about the neuronal circuit mechanisms (Amygdala – SNr

– DA), not only one particular brain area. For example, we propose that the amygdala operates on two distinct circuits within BG, with functional consequences manifest on both an immediate and a long-term time scale.

1) Amygdala→SNr→DA --- Learning in Environment

2) Amygdala→SNr→SC --- Action in Environment

So, we will continue studying this ‘suggested’ mechanism in the near future. To explain this, we revised the manuscript in the Discussion sections (Lines 298-324) and in the Result section (Lines 140-165) and modified Fig. 4c.

References

- Tepper, J. M., Martin, L. P. & Anderson, D. R. GABAA receptor-mediated inhibition of rat substantia nigra dopaminergic neurons by pars reticulata projection neurons. *J. Neurosci.* 15, 3092–3103 (1995).
- Kim, H. F., Ghazizadeh, A. & Hikosaka, O. Dopamine Neurons Encoding Long-Term Memory of Object Value for Habitual Behavior. *Cell* 163, 1165–1175 (2015).

The authors’ second major conclusion is that the amygdala acts on DA neurons through SNr. (Indeed, in the discussion, they say “we have found that tonic activity changes in DA neurons are generated by the amygdala-derived indirect input through SNr”). But there’s no direct evidence of this. Any other multi-synaptic path from amygdala to DA neurons could explain their findings. To make this conclusion, the authors would need to inhibit SNr while recording the effect of amygdala stimulation on DA neurons. They did not do this. At the very least, the authors need to tone down their conclusions. Their results may suggest that this disinhibitory process is possible (which we already knew from papers such as Steinberg et al., 2020, Neuron), but it certainly does not confirm that it happens.

Reply:

We agree with reviewer’s suggestions. Therefore, we have changed the initial part of the Discussion as “our data suggest that tonic activity changes in DA neurons are generated by the amygdala-derived indirect input through SNr.” Also, the Reviewer’s suggestion (“inhibit SNr while recording the effect of amygdala stimulation on DA neurons”) is very important. But there are at least two difficulties to perform this experiment. First, it is difficult to inhibit SNr neurons selectively because many DA neurons are often located among SNr neurons (Haber et al., *J. Comp. Neurol*, 1995). Second, the inhibition of SNr neurons causes a very strong behavioral effect: unexpected and repeated saccades (mostly to the contralateral side) (Hikosaka and Wurtz, *JNP*, 1985). This is because SNr neurons keep eye position stable by tonic inhibition on SC (superior colliculus) neurons and then generate a particular saccade by phasic disinhibition on SC neurons. Then, it is impossible to study the normal behaviors to study the function of DA neurons, including under particular environments. In contrast, the current process (AAV viral vector & O-stim) is useful to study the neuronal circuit mechanism to control behavior under environments.

To explain this, we added the following description in the Discussion.

(Lines 334- 339): To prove the idea, inhibiting SNr while recording the effect of amygdala stimulation on DA neurons is very important. However there are at least two difficulties to perform this experiment. First, it is difficult to inhibit SNr neurons selectively because many DA neurons are often located among SNr neurons (Haber et al., 1995). Second, the inhibition of SNr neurons causes a very strong behavioral effect: unexpected and repeated saccades (mostly to the contralateral side) (Hikosaka and Wurtz, 1985).

References

- Haber, S. N., Ryoo, H., Cox, C. & Lu, W. Subsets of midbrain dopaminergic neurons in monkeys are distinguished by different levels of mRNA for the dopamine transporter: comparison with the mRNA for the D2 receptor, tyrosine hydroxylase and calbindin immunoreactivity. *J. Comp. Neurol.* 362, 400–410 (1995).
- Hikosaka, O. & Wurtz, R. H. Modification of saccadic eye movements by GABA-related substances. II. Effects of muscimol in monkey substantia nigra pars reticulata. *J. Neurophysiol.* 53, 292–308 (1985).

• I am not convinced that the authors successfully identified DA neurons and separated them from non-DA neurons in the SNr. This calls into question all the rest of their findings. Figure S3 does not show a clear separation by baseline activity or spike duration. In addition, the authors appear to have required putative DA neurons to show phasic excitation to context or CS, and putative SNr neurons to show phasic inhibition to context. This is circular reasoning—the authors cannot identify these neurons based on their excitation or inhibition patterns, and then make independent claims about what these neurons do in the task. It seems quite possible that many of the neurons the authors claim are DA neurons are actually not DA neurons, which might explain the variability of recorded responses (with some neurons having longer-lasting responses than others). The authors should attempt to use other, more stringent, and non-circular methods to identify neurons, perhaps based on multiple features of the spike shape. Ultimately they may not be able to conclusively identify what neurons they are recording based on electrophysiology alone (this is a longstanding problem in the field; see, for example, Margolis ... Fields, *J Physiology* 2006). At the very least, the authors should use “putative DA” rather than “DA,” to make it clear that they have not conclusively identified these DA neurons. And perhaps they can look more closely at those neurons with baseline activity or spike duration that is on the border between SNr and DA. Do these neurons have particular patterns of task activity?

Reply:

As we described above, there are 2 groups of neurons in substantia nigra (SN). “Substantia nigra” in Wikipedia shows: “Although the substantia nigra appears as a continuous band in brain sections, anatomical studies have found that it actually consists of two parts with very different connections and functions: the pars compacta (SNpc) and the pars reticulata (SNpr). The pars compacta serves mainly as a projection to the basal ganglia circuit, supplying the striatum with dopamine. The pars reticulata conveys signals from the basal ganglia to numerous other brain structures. ... the neurons in pars reticulata are mainly GABAergic.”

Indeed, we have been studying these two groups (GABAergic in SNr, DA in SNc). One main study was to identify 2 groups of DA neurons (Matsumoto Hikosaka, *Nature* 2009). Even though

they have different functions (Value vs. Salience), we suggested that they were both DA neurons. To check this possibility, we examined various features of these groups of neurons. Then, we found that they have the same features in spike duration and basic firing rate (Fig S1 in Matsumoto Hikosaka, Nature 2009), which were different from the other group of neuron (GABAergic) in SNr.

Our current data in Fig S3 (previous version) are similar to those in Matsumoto Hikosaka, Nature 2009. However, we realized that there are some neurons that had a bit higher basic firing rates (> 10 sp/s). So, we removed them and generated a revised version for Fig S3 (which is equivalent to our previous data (Matsumoto Hikosaka, Nature 2009)). And we revised other figures according to using the new criteria. But there is no significant difference in our conclusion.

Reference

- Matsumoto, M. & Hikosaka, O. Two types of dopamine neuron distinctly convey positive and negative motivational signals. *Nature* 459, 837–841 (2009).

- The authors make a tidy distinction between “phasic” and “tonic” dopamine activity and argue that phasic activity is modulated by the conditioned stimulus (CS) and tonic activity is modulated by the environmental context. But I do not understand how the authors distinguish between phasic and tonic, and how they distinguish between CS and environment. When I look at the neural activity, I see very little difference between the response to the environmental stimulus and the response to the CS (see, for example, the green response vs the black response in Fig 1a and Fig 5g). It seems to me that there are longer and shorter responses to both the environment stimulus and the CS. How do the authors define “phasic” vs “tonic”? Isn’t the environmental context just another conditioned stimulus?
- The authors can do a better job explaining what is novel and significant in this manuscript. In particular, what does this paper add to our understanding of the links between CeA and substantia nigra DA neurons, that isn’t already in Steinberg et al., 2020?

Reply:

Here is our suggestion based on Reviewer’s questions (which we feel very important). In our task procedure, each trial started with a scene image which indicates an environment: possible reward (PR) or no reward (NR). In Amygdala, SNr, and DA, many neurons showed tonic activity in response to PR-environment, as shown in Fig 5a, 5d, 5g. Interestingly (as Reviewer described), the tonic activity often continued even after CS appeared until the end of trial (US). This may raise the question: Is the response to CS also tonic (not phasic)?

We then investigated this possibility in detail. After CS appeared, the tonic activity continued because the scene image (e.g., PR-environment) remained until the end of the trial. This suggests that the tonic activity is based on PR-environment, not CS. In addition, these neurons (Amygdala, SNr, DA) changed their activities phasically after CS appeared.

Amygdala: 100% > 50% > 0%

SNr: 0% > 50% > 100%

DA: 100% > 50% > 0%

These data suggest that Amygdala – SNr – SC circuit is controlled by PR-environment with tonic activity and by reward-prediction by CS.

However, the tonic activity almost disappeared when CS indicated 0% reward (as shown in Fig 5a, 5d, 5g). How does this happen? This makes Reviewer's question important (for us). We now have 1 suggestion: A specific CS (0% reward) abolishes the tonic activity based on PR-environment. This is actually very useful for real life. We are thus speculating the underlying mechanism. For example, this might be caused by the loss of the tonic activity in PR-environment which occurred in Amygdala, SNr, and eventually in DA. We hope we can publish any experimental data in near future.

We added the following description as well in the Results section.

(Lines 192-198): Then, after CS appeared, the tonic activity continued because the scene image (e.g., PR-environment) remained until the end of the trial. This suggests that the tonic activity is based on PR-environment, not CS. In addition, these neurons (Amygdala, putative SNr, putative DA) changed their activities phasically after CS appeared (Amygdala: 100% > 50% > 0%, SNr: 0% > 50% > 100%, DA: 100% > 50% > 0%), These data suggest that Amygdala – SNr – SC circuit is controlled by PR-environment with tonic activity and by reward-prediction by CS. However, the tonic activity almost disappeared when CS indicated 0% reward (as shown in Fig 5a, 5d, 5g).

Minor comments:

- Why are the measured latencies, even in the amygdala neurons, so long? Previous work combining optogenetics and electrophysiology often report latencies less than 5 or 10ms; longer latencies were assumed to be polysynaptic.

Reply:

We agree with this. Polysynaptic is one of the possibilities because depending on the place where the stimulation in the cell soma, the response latency was changed. According to a previous paper that showed optogenetic stimulation effects by in-vitro experiments in which effects of the stronger activation and the shorter latency could be seen when stimulating through the closest optrode right under the electrode. But, even slightly 100µm away from the center of the soma, the latency was delayed (Welkenhuysen, M. et al. . Sci Rep (2016)). Unlike the in-vitro optogenetic stimulation, in our in-vivo stimulation in the primate, we thought that recording the wide variety of neuronal effects could be essential to estimate neuronal circuits in natural circumstances in vivo. Then we used a linear probe in which the distance range from the optic fiber to an electrode channel was around a minimum 100 µm to a maximum of 2.5 mm away, which could include direct effects as well as polysynaptic effects that showing different types of inhibition and excitation by optogenetic stimulation of ChR2 expressing neurons.

- The sample sizes are unclear throughout. How does the example in Fig. 1 generalize to all other recorded neurons, and how many recorded neurons were there? For Figs. 3-5, please specify sample size for the statistics.

Reply:

The population of neurons tested by optogenetics and neurons recorded while performing the task overlaps only partially due to the difficulties of holding neurons during the stimulation and the task. Therefore, it turned out that the overall number of neurons would not be high. However, we have confirmed that some neurons have congruent excitatory or inhibitory responses both during the task and to the O-stim (Fig. S6).

- Fig. 4c is overly speculative. Why is there a converging input from multiple CeA neurons to one SNr neuron? Another explanation for the results in e and h is that as the stimulation site is moved away from the recorded neuron, the light reaching the terminals will be dimmer and therefore less reliable in driving spikes. The data also does not directly support the projection from the recorded SNr neuron to the DA neurons. Finally, the title of Fig. 4 is confusing. How does the data in this figure relate to 'environmental information'?

Reply:

The original version seemed to suggest that many CeA neurons project to 1 particular SNr neuron, which cannot be suggested (as the Reviewer indicated). We then realized that it is important to discuss and predict neuronal activities based on various conditions. So, we revised Fig. 4c by including several groups of synaptic connections: 1) Inputs from different groups of CeA neurons (#5 compared with #4 in Fig. 4c), 2) Inputs from the same groups of CeA neurons (#2 compared with #4 in Fig. 4c). Then, O-stim at 1 position (e.g., #1-7) would affect (inhibition or excitation) neurons close to the O-stim location, because O-stim would modulate many synaptic inputs to these neurons (e.g., neuron close to #2, #4, #5).

These effects occurred strongly in SNr neurons (Fig. 3d), suggesting that they received direct inputs from CeA neurons. This is also supported by other experimental data: If the SNr neuron is closer to O-stim, the effect of O-stim was quicker (Fig. 4c) and stronger (Fig. 4h).

In contrast, DA neurons showed none of these effects: Neither latency (Fig. 4f) nor strength (Fig. 4i) was related to the distance from O-stim. These data suggest that DA neurons are less likely to receive synaptic inputs directly from CeA (unlike SNr neurons). Then, it is likely that DA neurons then receive synaptic inputs indirectly from CeA. It is then likely that the indirect inputs are mediated by SNr neurons. Such a connection (SNr to DA) has been shown anatomically (Uchida et. al., 2012).

Then, we are now proposing the detail of the synaptic input (CeA-SNr-DA) (Fig. 4c). If DA neurons receive inputs from SNr neurons nearby, the effect of O-stim would be slightly similar to SNr neurons (which we have not observed). Then, DA neurons are likely to receive inputs from SNr neurons in various positions. We are now wondering if this is important for behavioral functions (maybe for future experiments).

Reference

- Watabe-Uchida, M., Zhu, L., Ogawa, S. K., Vamanrao, A. & Uchida, N. Whole-brain mapping of direct inputs to midbrain dopamine neurons. *Neuron* 74, 858–873 (2012).

- There is no data supporting the statement (in the abstract and last sentence of the manuscript) that 'the amygdala provides additional source of learning to the BG circuit'. Did the neuronal

response to the scene image develop over learning? Did optical stimulation of CeA affect learning?

Reply:

We agreed with this comment. The abstract and last sentence of the manuscript was changed, as shown below.

“the amygdala may provide an additional source of learning to BG circuits, namely contingencies imposed by the environment.”

• Fig. 5 shows the neural responses to the PR environment. How does the neural activity look like in the NR environment? Are SNr and amygdala neurons tuned to either PR or NR environment specifically, as the example DA neuron shown in Fig. 1?

Reply:

As the suggestion, these neurons we showed in the paper were tuned to PR environment specifically. The data in NR environment was shown in “Fig. R1,” which is attached to the bottom of this document.

• Figure 1: are trials in the green of the raster plots sorted in the same way as in the grey-black part? The sudden jump from one to three panels is a bit confusing.

Reply:

The order of the raster plots is sorted by the trial order in each group of the raster plots. We made the separation clearer for 100 %, 50 %, and 0 %, respectively, and put the texts for those plots (Figure 1).

• Line 84-85: ‘the absence in the environmental context’ is confusing. There is always a context for any form of task. Perhaps the authors mean in the absence of a context-predicting cue?

Reply:

We modified the text as this suggestion (line 82). Thank you for your comments.

• Fig. S1c-d: It would be good to show the neural response to optical stimulation in a ChR2+ animal.

Reply:

Thank you for your comments. We put the data in Fig. S1.

• Fig 3: typo in the y-axis of b, d, and f

Reply:

Thank you. We have corrected this.

Reviewer #3 (Remarks to the Author):

Major issues:

1. The authors use aerial photographs for contextual cues. There are two issues. First, it is very unlikely that monkeys can understand the meaning of the pictures – whether these pictures are “scenes” or representing an “environment”. It is most likely that these images are nonsense images to monkeys. Second, because of this reason, the difference between the contextual cues and conditioned stimuli (fractals) are mainly the size in the computer monitor. Therefore, I am not very convinced that the contextual cues had a special meaning that qualifies to be a “scene” or “environment” beyond providing broader task information in the task. Because of these reasons, whether the contextual cues and the CSs had fundamental differences that are processed by different neural circuits remains largely unclear. It remains unclear whether contextual cues and CSs of a same size can cause different firing patterns of the neurons studies in this work.

Reply:

This is a very important question for us, because animals (including humans) may have difficulty in discriminating visual objects, especially if they have complex features. Relatively recently, however, we found unexpected mechanisms of visual discrimination in monkeys (Yasuda et al., *J Neurosci*, 2012). Here, we tested if the animal as well as brain areas (e.g., SNr) can remember the value of each object for a long time. We then found that neurons in the posterior part of the basal ganglia (e.g., cdlSNr) can discriminate the values of so many objects ($n = 280$) for a long time (> 100 days). We then tested many monkeys and found that all of them learned many complex objects ($n > 300$) and remember the value of each object for a long time (> 1 year). They were also able to identify each object for a long time, even when it was in a peripheral position (e.g., 15 degrees away from center) (Ghazizadeh et al., *J Vision* 2016).

These data then raised another question: Is Environment useful to predict upcoming objects. Then, we generated many Environments (or Scenes) in which various objects are hidden and sometimes appear. In a previous study (Maeda et al., *PLoS Biol*, 2018) we generated many Environments (e.g., 140 objects are separated selectively or combined differently in 56 Environments) (Fig 1c in Maeda et al., *PLoS Biol*, 2018). To make such many Environments, we needed to use complex visual features for each Environment. We then wondered if the subject (monkey) can discriminate such complex Environments (as Reviewer asked). But we found that all subjects and many neurons in Amygdala discriminate such Environments quickly and remember them for a long time. When 2 different Environments indicated the changes of the same objects (e.g., Object-A $>$ Object-B in Environment-X, Object-B $>$ Object-A in Environment-Y), the subject and neurons can change their choices very quickly (Kunimatsu et al., *PNAS*, 2021).

Based on these unexpected data, we are now thinking about the neuronal mechanism of the discrimination and memory of complex visual objects. We suspect that such ability is not selective to monkeys; other animals (e.g., rat) have the same activities. This is probably necessary because they often navigate through various areas with complex features in order to find good objects, friends, risky animals etc. We hope that other groups of researchers can use complex visual objects to study their future research.

References

- Yasuda, M., Yamamoto, S. & Hikosaka, O. Robust representation of stable object values in the oculomotor Basal Ganglia. *J. Neurosci.* 32, 16917–16932 (2012).

- Ghazizadeh, A., Griggs, W. & Hikosaka, O. Object-finding skill created by repeated reward experience. *Journal of Vision* 16, 17–13 (2016).
- Maeda, K., Kunimatsu, J. & Hikosaka, O. Amygdala activity for the modulation of goal-directed behavior in emotional contexts. *PLoS Biol.* 16, e2005339 (2018).
- Kunimatsu, J., Yamamoto, S., Maeda, K. & Hikosaka, O. Environment-based object values learned by local network in the striatum tail. *Proc. Natl. Acad. Sci. U.S.A.* 118, e2013623118 (2021).

2. Although the tonic firing patterns are somewhat similar (or roughly mirror-image) between amygdala, SNr and dopamine neurons, their firing patterns have important differences. For example, dopamine neurons increased their tonic firing after the timing cue (Fig. 5g). Amygdala neurons have a weak but similar tendency, but SNr neurons did not change the tonic level after the timing cue. The pattern of phasic responses were also different. The authors use optogenetics to infer synaptic connectivity but have not performed manipulations during the task. Furthermore, it is unclear whether the amygdala neurons that the authors recorded indeed project to the SNr. Considering these limitations, the conclusion that disinhibition via SNr generates tonic modulations in dopamine neurons (which is in the title) appears to be premature.

Reply:

Each of the 3 groups of neurons (Amygdala, SNr, DA) showed 2 kinds of activity (Phasic and Tonic) in response to Environment, especially Reward-predicting (RP). In the preceding figures (Fig 5, old Fig S4), it was not so clear how (and if) such groups of activity are sent through the circuit (Amygdala – SNr – DA). We thus made changes in these data (Fig 5, new Fig S4) by separating to 3 groups: Phasic(+) & Tonic(+) (Fig 5), Phasic(+) only (Fig S4-left), Phasic(+) & Tonic(-) (Fig S4-right). In Fig 5 the discrimination between Phasic and Tonic activities is not so clear because they have the same orientations: Amygdala = Phasic Excitation followed by Tonic Excitation (similar sizes), SNr = Phasic Inhibition followed by Tonic Inhibition (similar sizes), DA = Phasic Excitation followed by Tonic Excitation (Phasic > Tonic). In Fig S4, however, the discrimination between Phasic and Tonic activities is very clear because Tonic activity is absent (Fig S4-left) or opposite (and small) (Fig. S4-right). We hope we can examine which neurons are actually work as the real neuronal circuit.

3. The authors do not report response patterns of dopamine, SNr nor amygdala neurons. Were firing patterns of these neurons consistent with the author's model?

Reply:

We also generated a figure to show responses in NR contexts in addition to PR contexts. The data are shown in "Fig. R1," which is attached to the bottom of this document. According to these population responses in PR contexts and NR contexts, when amygdala neurons showed strong responses, SNr neurons were inhibited, and DA neurons were excited, which firing patterns consistent with our model.

Figure R1

REVIEWER COMMENTS

Reviewer #1 (Remarks to the Author):

My comment regarding the GABAergic neurons has been dealt with rather 'cosmetically' by changing nomenclature to 'Str-like' neurons. But the idea is that GABAergic neurons are embedded in the DA cells and in the SNr may not have such a sharp line between them and that functional /electrophysiologic responses could be similar. That is important because of effects on DA subpopulations.

I would suggest just adding a sentence saying that in the primate, GABAergic neurons in SNc are found (they make up 1/3 of all neurons in the SNc), and that they merge with those in the SNr. Then, just refer to recorded GABA-like neurons as 'putative GABAergic neurons' (rather than localizing to the SNr) . Unless there is good reason to believe that most are isolated in the SNr, this makes the most conceptual sense.

Reviewer #2 (Remarks to the Author):

The authors have responded to many of my prior reservations, but I think more can be done to align the conclusions with the data and eliminate possible confounds.

First, I think the title and abstract should be revised. The data simply do not show that dopamine neurons are activated in reward environments by disinhibition from the amygdala. This mechanism is possible, but was not directly tested. At the very least, the authors could change "activated" to "may be activated", or change the title to something such as, "Putative amygdala circuit for context-dependent activation of DA neurons". Similarly, in the abstract, the authors should add "may" before "regulate putative dopamine" and change "These responses are mediated by" to "These responses may be mediated by." All of these declarative statements are not backed up by the presented data, and thus have the potential to mislead readers.

I am still struck by the circularity of the neuron identification. Many of the paper's conclusions rely on the mirror-image responses of amygdala, SNr, and dopamine neurons, but it seems like these responses were required to even include the neurons in the paper. How many putative dopamine neurons were removed from analysis because they did not show excitation to 100% reward CS, how many SNr neurons were excluded because they did not show inhibition to the scene image, and how many amygdala neurons were excluded because they did not show excitation to the scene image? What happens if you include these neurons?

Finally, there are two distinctions that I would like to see discussed in the manuscript. One is between "phasic" and "tonic". How do the authors define these terms? The second is between "context" and "CS"? What makes one stimulus a "context" stimulus and another stimulus a "CS"? If anything, the fact that the 0% CS can abolish the 'context' response makes me question the distinction between CS and context.

POINT-BY-POINT RESPONSE TO COMMENTS RAISED BY REVIEWERS

We would like to thank the reviewers for careful and thorough reading of our revised manuscript and for invaluable comments and constructive suggestions, which greatly helps to improve the quality of the manuscript. In line with the comments and suggestions raised by the reviewers, we have revised our manuscript as completely as possible.

We respond below in detail to each of the reviewer's comments. We hope that the reviewers will find our responses to their comments satisfactory.

Please find below the reviewers' comments repeated in black and our responses (blue) inserted after each comment. To facilitate the work of the reviewers, we provide the manuscript in which the changes are shown as track changes in a file "Manuscript_tracked changes.docx". Line numbers shown below are confirmed with a final version of the PDF file "Manuscript.pdf".

We look forward to hearing from you at your earliest convenience.

Sincerely yours,

Kazutaka Maeda and Okihide Hikosaka

Your revision should address all the points raised by our reviewers (see their reports below). Specifically, we would like to highlight the need to tone down several claims of the paper: please only make conclusions that are supported by the data, and clearly distinguish between what the data show versus what the data may suggest - especially in terms of the suggested mechanism of dopamine neuron disinhibition via the amygdala.

Also note that while unfortunately, Reviewer #3 wasn't able to comment on this manuscript version, we asked one of the other reviewers to comment on how Reviewer #3's prior concerns were addressed. Their advice echoed the need to tone down the claims of the paper as mentioned in Reviewer #3's original report. Further, the collective reviewer feedback emphasized the need to better clarify the distinction between contextual stimuli and the CS, as it still remains unclear whether there is a qualitative difference in how the contextual scene and the CS are experienced, so that only the scene and not the CS engages the amygdala.

(Reply): Thank you for asking another reviewer to comment on our responses to the concerns raised by Reviewer #3. Because these were similar to the concerns raised by Reviewer #2, we describe a response at the last part of the responses to the comments raised by Reviewer #2.

When resubmitting, you must provide a point-by-point response to the reviewers' comments. Please show all changes in the manuscript text file with track changes or color highlighting. If you are unable to address specific reviewer requests or find any points invalid, please explain why in the point-by-point response.

REVIEWER COMMENTS

Reviewer #1 (Remarks to the Author):

My comment regarding the GABAergic neurons has been dealt with rather 'cosmetically' by changing nomenclature to 'Str-like' neurons. But the idea is that GABAergic neurons embedded in the DA cells and in the SNr may not have such a sharp line between them and that functional /electrophysiologic responses could be similar. That is important because of effects on DA subpopulations.

I would suggest just adding a sentence saying that in the primate, GABAergic neurons in SNc are found (they make up 1/3 of all neurons in the SNc), and that they merge with those in the SNr. Then, just refer to recorded GABA-like neurons as 'putative GABAergic neurons' (rather than localizing to the SNr). Unless there is good reason to believe that most are isolated in the SNr,

this makes the most conceptual sense.

(Reply):

As the reviewer pointed out, we could not clearly determine whether putative GABAergic neurons recorded were located in SNc or SNr. Although it has been reported that 1/3 of the SNc neurons are GABAergic in the rat (Nair-Roberts et al., 2008), such neurons were not so many or few in the monkey (Smith et al., 1987).

Another reason for difficulties in regional determination is the complexity of the border between SNc and SNr. In general, DA neurons in SNc are located dorsally, and GABAergic neurons in SNr are located more ventrally. However, they are not completely separated to make their discrimination very hard, as shown in figures below (Kawagoe et al., 2004, Sato and Hikosaka, 2002).

According to the reviewer's suggestion, we have therefore decided to change "SNr neuron" to "GABAergic neuron in SN" throughout the entire manuscript and figures.

We have also added the following descriptions in the manuscript as reviewer's suggestion (lines 199-203).

"The recorded putative GABAergic neurons were found in the dorsal part as well as in the ventral part of SN (Fig. 5f). The dorsal part may be included in SNc that is primarily composed of DA neurons, but contains a small number of GABAergic neurons (Smith et al., 1987; Nair-Roberts et al., 2008). In this study, we analyzed those as a group of putative GABAergic neurons."

FIG. 8. Recording sites of DA neurons in monkey G (left) and monkey M (right). Three coronal sections are shown for monkey G (A-C) and for monkey M (H-J), rostrocaudally with 1-mm intervals. DA neurons that were and were not fully examined are shown by filled and open circles, respectively. Horizontal bars indicate electrolytic marks. Photomicrographs in D and F (Nissl-stained sections) represent parts of C (monkey G) and J (monkey M), respectively (indicated by dashed circles). TH-stained sections in E and G were adjacent to the sections in D and F, respectively, and their positions correspond to the dashed rectangles in D and F. Aggregation of TH-positive cell bodies and dendrites are visible. SNc, substantia nigra pars compacta; SNr, substantia nigra pars reticulata; STN, subthalamic nucleus.

(Kawagoe et al., 2004)

Figure 10. Recording sites of SNr neurons. *A*, A coronal histological section showing electrode tracks aimed at the SNr. In the enlarged photograph (*B*) are indicated the SNr and the substantia nigra pars compacta (SNc). An electrolytic mark (arrow) in the dorsolateral part of the SNr indicates the location where the neuron shown in Figure 3 was recorded. *C-H*, Recording sites of SNr neurons mapped on coronal histological sections from rostral (*C*) to caudal (*H*) with 0.5 mm intervals. The decreasing and increasing types (for their post-cue activity) are indicated by filled and open circles, respectively. Small dots indicate neurons that were not related to IDR or ADR.

(Sato and Hikosaka, 2002)

(References)

Nair-Roberts, R. G. et al. Stereological estimates of dopaminergic, GABAergic and glutamatergic neurons in the ventral tegmental area, substantia nigra and retrorubral field in the rat. *Neuroscience* 152, 1024–1031 (2008).

Smith, Y., Parent, A., Seguela, P. & Descarries, L. Distribution of GABA-immunoreactive neurons in the basal ganglia of the squirrel monkey (*Saimiri sciureus*). *J Comp Neurol* 259, 50–64 (1987).

Kawagoe, R., Takikawa, Y. & Hikosaka, O. Reward-predicting activity of dopamine and caudate neurons—a possible mechanism of motivational control of saccadic eye movement. *J. Neurophysiol.* 91, 1013–1024 (2004).

Sato, M. & Hikosaka, O. Role of Primate Substantia Nigra Pars Reticulata in Reward-Oriented Saccadic Eye Movement. *J Neurosci* 22, 2363–2373 (2002).

Reviewer #2 (Remarks to the Author):

The authors have responded to many of my prior reservations, but I think more can be done to align the conclusions with the data and eliminate possible confounds.

First, I think the title and abstract should be revised. The data simply do not show that dopamine neurons are activated in reward environments by disinhibition from the amygdala. This mechanism is possible, but was not directly tested. At the very least, the authors could change “activated” to “may be activated”, or change the title to something such as, “Putative amygdala circuit for context-dependent activation of DA neurons”. Similarly, in the abstract, the authors should add “may” before “regulate putative dopamine” and change “These responses are mediated by” to “These responses may be mediated by.” All of these declarative statements are not backed up by the presented data, and thus have the potential to mislead readers.

(Reply):

Based on the reviewer’s suggestion, we have changed the title of this paper to "Environmental context-dependent activation of dopamine neurons via putative amygdala-nigra pathway". We have further revised the Abstract in line with the reviewer’s advice: “may regulate putative dopamine” and “These responses may be mediated by” (lines 23 and 26).

I am still struck by the circularity of the neuron identification. Many of the paper’s conclusions rely on the mirror-image responses of amygdala, SNr, and dopamine neurons, but it seems like these responses were required to even include the neurons in the paper. How many putative dopamine neurons were removed from analysis because they did not show excitation to 100% reward CS, how many SNr neurons were excluded because they did not show inhibition to the scene image, and how many amygdala neurons were excluded because they did not show excitation to the scene image? What happens if you include these neurons?

(Reply):

Some descriptions to discriminate putative dopamine and GABAergic neurons have been added to the Results (lines 112-119 and 447-457).

According to the whole neuron recording data, out of the nigral neurons ($n = 386$) that had less than 10 sp/s baseline activity, 81 neurons (20.98 %) met the criteria as putative dopamine neurons. On the other hand, out of the nigral neurons ($n = 158$) with more than 15 sp/s baseline activity, 31 neurons (19.62 %) were regarded as putative GABAergic neurons.

Out of all recorded neurons ($n = 324$) in the amygdala, 137 neurons (42.68 %) showed excitation to scene images.

Below show the numbers of neurons when each of the three populations was classified by tonic responses to scene images.

1. Baseline FRs (<10 sp/s) in SN (n=386): Excitation (n=95), Inhibition (n=33), No modulation (n=258)
2. Baseline FRs (>15 sp/s) in SN (n=158): Excitation (n=60), Inhibition (n=35), No modulation (n=63)
3. Amygdala (n=324): Excitation (n=83), Inhibition (n=52), No modulation (n=189)

The results on groups 1 and 3 are consistent with the results we analyzed in the neuronal populations classified into putative dopamine neurons and amygdala neurons as described above, respectively.

Group 2 includes GABAergic neurons we classified in this study. These GABAergic neurons contained a larger number of inhibition-type neurons (Fig. 5d,e). However, neurons other than GABAergic neurons in group 2 were mostly of excitation type. In this study, we did not further analyze the data on higher-frequency excitatory responses in SN due to the limitation of identifying specific cell types based on physiological recordings. In a future study, we would like to determine the identity and function of such excitatory responses in SN.

Finally, there are two distinctions that I would like to see discussed in the manuscript. One is between “phasic” and “tonic”. How do the authors define these terms? The second is between “context” and “CS”? What makes one stimulus a “context” stimulus and another stimulus a “CS”? If anything, the fact that the 0% CS can abolish the ‘context’ response makes me question the distinction between CS and context.

(Reply):

Our studies suggest that both phasic and tonic activities are important for the goal-directed behavior, especially when the outcome occurs after a sequence of events within a particular environment. Tonic activity across the sequence indicates the final outcome (e.g., good or bad). According to our data, this activity in fact emerged in the amygdala, which is sent to DA neurons through GABAergic neurons (by disinhibition). In addition, it would also be important to evaluate each step of sequential action, since individual steps may not be maintained consistently. Then, a phasic response is also important to evaluate each step. This actually occurred in DA neurons because they showed phasic activity strongly (often with tonic activity) (Figs. 1, 5, S4). Then, such a phasic response is very important to evaluate each step of sequential action.

As the reviewer pointed out, a “context” stimulus is a kind of CSs. In this study, however, we use the term “context” to distinguish from visual (fractal) CSs, as many previous works used to find dopamine neurons in the primate SNc (Schultz et. al., 1997 and Matsumoto and Hikosaka 2009). These CSs are primarily small and may also subserve as saccadic target cues. Actually, animals made a saccade to fractal CSs when it appeared. Here we presented a scene image first as contextual information before appearing the fractal CSs that is directly associated with reward or punishment. Such information changes brain activities and animal behaviors because the animals can prepare following events (CSs) while the scene appears. The behavioral changes due to contextual information are different from other behaviors to target cues like fractal CSs. These data have been demonstrated in our previous paper (Maeda et al., 2018), and the proportions of reaction time changes depending on the contextual information and target information are also clearly discriminated and fit with the Linear Approach to Threshold with Ergodic Rate model (Noorani and Carpenter, 2016).

(References)

Schultz, W., Dayan, P. & Montague, P. R. A neural substrate of prediction and reward. *Science* 275, 1593–1599 (1997).

Matsumoto, M. & Hikosaka, O. Two types of dopamine neuron distinctly convey positive and negative motivational signals. *Nature* 459, 837–841 (2009).

Maeda, K., Kunitatsu, J. & Hikosaka, O. Amygdala activity for the modulation of goal-directed behavior in emotional contexts. *Plos Biol* 16, e2005339 (2018).

Noorani, I. & Carpenter, R. H. S. The LATER model of reaction time and decision. *Neurosci Biobehav Rev* 64, 229–251 (2016).

REVIEWERS' COMMENTS

Reviewer #2 (Remarks to the Author):

I believe the authors responded adequately to all of my concerns, and I only have two minor remaining suggestions. I would still like a sentence somewhere in the manuscript that defines what the authors mean by "phasic" and "tonic" -- is there a cutoff in terms of number of seconds? These are terms that are often used in the literature to mean different things and I think the authors have the chance to make a helpful contribution by providing a concrete definition. Second, I think the authors made a reasonable argument in their rebuttal that a context is different from a CS because the animals behave differently in response to the two types of stimuli. However, the authors did not include any statement as such in the actual manuscript. They included these arguments only in the rebuttal, not in the manuscript itself. I think it would strengthen the manuscript, and clear up any confusion, if the authors include a sentence acknowledging that a context is a type of CS, but that the two stimuli can be distinguished through the behaviors they elicit from the animals.

POINT-BY-POINT RESPONSE TO COMMENTS RAISED BY REVIEWERS

We would like to thank the reviewers for carefully reading a revised manuscript and for providing us with invaluable comments, which greatly helps to improve the quality of our work. Please find below the comments raised by Reviewer #2 (in black) and our responses to them (in blue). We hope that we have now revised the manuscript satisfactorily in line with the reviewer's comments, and that the revised manuscript will be accepted as it is.

We look forward to hearing from you at your earliest convenience.

Sincerely yours,

Kazutaka Maeda

Reviewer #2 (Remarks to the Author):

I believe the authors responded adequately to all of my concerns, and I only have two minor remaining suggestions. I would still like a sentence somewhere in the manuscript that defines what the authors mean by "phasic" and "tonic" -- is there a cutoff in terms of number of seconds? These are terms that are often used in the literature to mean different things and I think the authors have the chance to make a helpful contribution by providing a concrete definition.

According to this suggestion, we have added some descriptions to the Discussion section as follows (see lines 287-295):

“One of the critical differences between tonic and phasic activity changes is how such responses continue during the period of an event. Since the tonic activity changes occur in a linear, ramp-like manner throughout the event, it is important to sustain the information about the current status until it changes. On the other hand, even if the phasic activity does not maintain response changes throughout the event, it seems critical to evaluate each step of a sequential action as individual steps may not be maintained consistently. From this viewpoint, it may not be plausible to discriminate simply the definitions of tonic and phasic activity changes just by the difference in the response duration. Yet, the tonic activity should maintain response changes during the period of the event.”

Second, I think the authors made a reasonable argument in their rebuttal that a context is different from a CS because the animals behave differently in response to the two types of stimuli. However, the authors did not include any statement as such in the actual manuscript. They included these arguments only in the rebuttal, not in the manuscript itself. I think it would strengthen the manuscript, and clear up any confusion, if the authors include a sentence

acknowledging that a context is a type of CS, but that the two stimuli can be distinguished through the behaviors they elicit from the animals.

According to this suggestion, we have added some descriptions to the Discussion section as follows (see lines 265-277):

“In this study, we used large visual scene images to represent environments that predict different emotional outcomes (e.g., reward). These scene images can be distinguished from fractal CSs which have been utilized to find dopamine neurons in the primate SNc in many previous works^{21,22}. The fractal CSs are primarily small and may also subserve as saccadic target cues. In our experimental task, a large scene image first appears as environmental information before appearing the fractal CSs that are directly associated with reward or punishment. Such information changes neuronal activities and animal behaviors, because the animals may prepare for subsequent events related to the CSs while the scene image appears. In fact, the behavioral changes based on the environmental information are different from those according to target cues such as fractal CSs, as demonstrated in our prior study⁶. In this paper, we clearly discriminated the proportions of reaction time changes depending on the environmental information and target information, and considered that this fits the Linear Approach to Threshold with Ergodic Rate model³⁹.”